# Old orogen – young topography: lithological contrasts controlling erosion and relief formation in the Bohemian Massif

Jörg Robl[1], Fabian Dremel[1], Kurt Stüwe[2], Stefan Hergarten[3], Christoph von Hagke[1], and Derek Fabel[4]

[1]Department of Environment and Biodiversity, Division of Geology and Physical Geography, University of Salzburg, 5020 Salzburg, Austria
[2]Institute for Earth Sciences, University of Graz, 8020 Graz, Austria
[3]Institute of Earth and Environmental Sciences, Albert-Ludwigs-Universität Freiburg, Freiburg, Germany
[4]Scottish Universities Environmental Research Center, The University of Glasgow, East Kilbride, United Kingdom

**Correspondence:** Jörg Robl (joerg.robl@plus.ac.at)

**Abstract.** In several low mountain ranges throughout Europe, high-grade metamorphic and granitic rocks of the Variscan orogen are exposed – even though the topography of this Paleozoic mountain range was largely leveled during the Permian and later covered by sediments. The Bohemian Massif is one of these low mountain ranges and consists of high-grade metamorphic and magmatic rocks that dip southward below the weakly consolidated Neogene sediments of the Alpine Molasse Basin. Morphologically, the Bohemian Massif is characterized by rolling hills and extensive low-relief surfaces above 500 m, which contrast with deeply incised canyons characterized by steep and morphologically active valley flanks. These morphological features and the occurrence of marine sediments several hundred meters above sea level are a clear indication of relief rejuvenation due to significant surface uplift during the last few million years.

To constrain landscape change and its rate, we used the concentration of cosmogenic $^{10}$Be in river sands to determine 20 catchment-wide erosion rates and correlated these with topographic metrics characterizing both the hillslopes and the drainage systems. Erosion rates range from 22 to 51 m per million years, which is generally low compared to tectonically active mountain ranges such as the Alps. Low erosion rates in the Bohemian Massif seem to contradict the steep topography observed close to the receiving streams (i.e., the Danube River and the Vltava River), which have morphological characteristics of alpine landscapes. We found that erosion rate is correlated with catchment-wide topographic metrics, representing both hillslope and channel morphology. Highest erosion rates occur in catchments featuring high channel steepness and a large area fraction with significant geophysical relief. Catchments with abundant deeply incised canyons erode about twice as fast as those characterized primarily by low-relief surfaces.

We interpret the measured erosion rates and related topographic patterns as the landscape response to slow and large-scale uplift in concert with strong variations in bedrock erodibility between rocks of the Bohemian Massif and the Neogene Molasse basin. We propose that lithology is ultimately responsible for the topographic difference between the mountainous Bohemian Massif and the low-relief Molasse zone despite a common uplift history during the last few million years. As erosion progresses, the basement rocks with their high resistance to erosion are exposed. The repeated emergence of such bedrock barriers reduces the erosion rate during topographic adjustment and governs the evolution of low-relief surfaces at different elevation levels.

The resulting stepped landscape requires neither spatial nor temporal changes in uplift rate but can form by erodibility contrasts
under uniform uplift conditions.

## 1   Introduction

The Bohemian Massif is one of several Variscan massifs in Europe with peak elevations exceeding 1.5 km (Olivetti et al., 2016;
Wetzlinger et al., 2023). Far from active plate boundaries and tens of kilometers north of the alpine deformation front, erosion-
resistant, metamorphic- and granitic rocks of the Variscan basement protrude from the soft Neogene sediments of the Molasse
Basin, an asymmetric flexural foreland basin of the Europen Alps (Fig. 1). The bedrock within the Bohemian Massif consists
predominantly of granitoids (South Bohemian batholith) and high-grade metamorphic rocks such as ortho- and paragneisses
and migmatites (Fig. 1b) (Wessely, 2006). The formation of these rocks can be traced back to the Variscan orogeny, which
took place between 340 and 300 million years ago (Franke, 2014; Kroner and Romer, 2013; O'Brien and Carswell, 1993). To
the south, rocks of the crystalline basement dip below the Neogene sediments of the Alpine Molasse Basin. The thickness of
the sediments increases southwards towards the Alps and reaches up to 4.5 km (e.g., Gusterhuber et al., 2012). This is due to
the asymmetry of the foreland basin that has been formed by the load of the Alpine orogen and the bending of the lithosphere
(e.g., Genser et al., 2007). Sandy and silty rocks of the Upper Marine Molasse (Ottnangian) are exposed over large areas with
isolated outcrops (e.g. Kobernaußerwald) of the stratigraphically higher Upper Freshwater Molasse (Pannonian). The wide
grain size spectrum (clay to gravel) of these sediments indicate lacustrine and fluvial depositional conditions (Baumann et al.,
2018).

Since the realm of the Bohemian Massif was largely leveled in the Permian (Bourgeois et al., 2007; Danišík et al., 2010;
Hejl et al., 1997, 2003; Ziegler and Dèzes, 2007), surface uplift and the emergence of the low mountain topography must have
occurred much later, namely after the deposition of Neogene sediments in the Molasse basin. Relics of Molasse sediments
can now be found in elevated low relief surfaces within the Bohemian Massif several hundred meters above sea level (e.g.,
Wessely, 2006). Slow uplift with low gradients in uplift rate since 8 - 10 Myr have affected large areas of Central and Eastern
Europe including the Bohemian Massif and the adjacent Molasse Basin (Baran et al., 2014; Genser et al., 2007; Gusterhuber
et al., 2012), but also large parts of the Eastern Alps and the Styrian Basin (Gradwohl et al., 2024; Legrain et al., 2015; Robl
et al., 2008, 2015; Wagner et al., 2010, 2011). Onset of uplift of the Bohemian Massif and inversion of the Molasse Basin
from a depositional environment to an eroding landscape are not well constrained but a major erosional phase started at 8 Myr
and slowed down at 4 Myr (e.g., Gusterhuber et al., 2012). The strata of the Molasse Basin north of the alpine deformation
front are largely undisturbed but slightly tilted towards the west, which indicates a minor west-east gradient in the uplift rate
(Gusterhuber et al., 2012). An average surface uplift rate of about 100 to 150 $\mathrm{m\,Myr^{-1}}$ can be estimated from peak elevations
of the Molasse zone with about 800 m at the Kobernaußerwald–Hausruck region (Baumann et al., 2018) and indications that
at least 450 m of sediment have already been eroded (e.g., Gusterhuber et al., 2012).

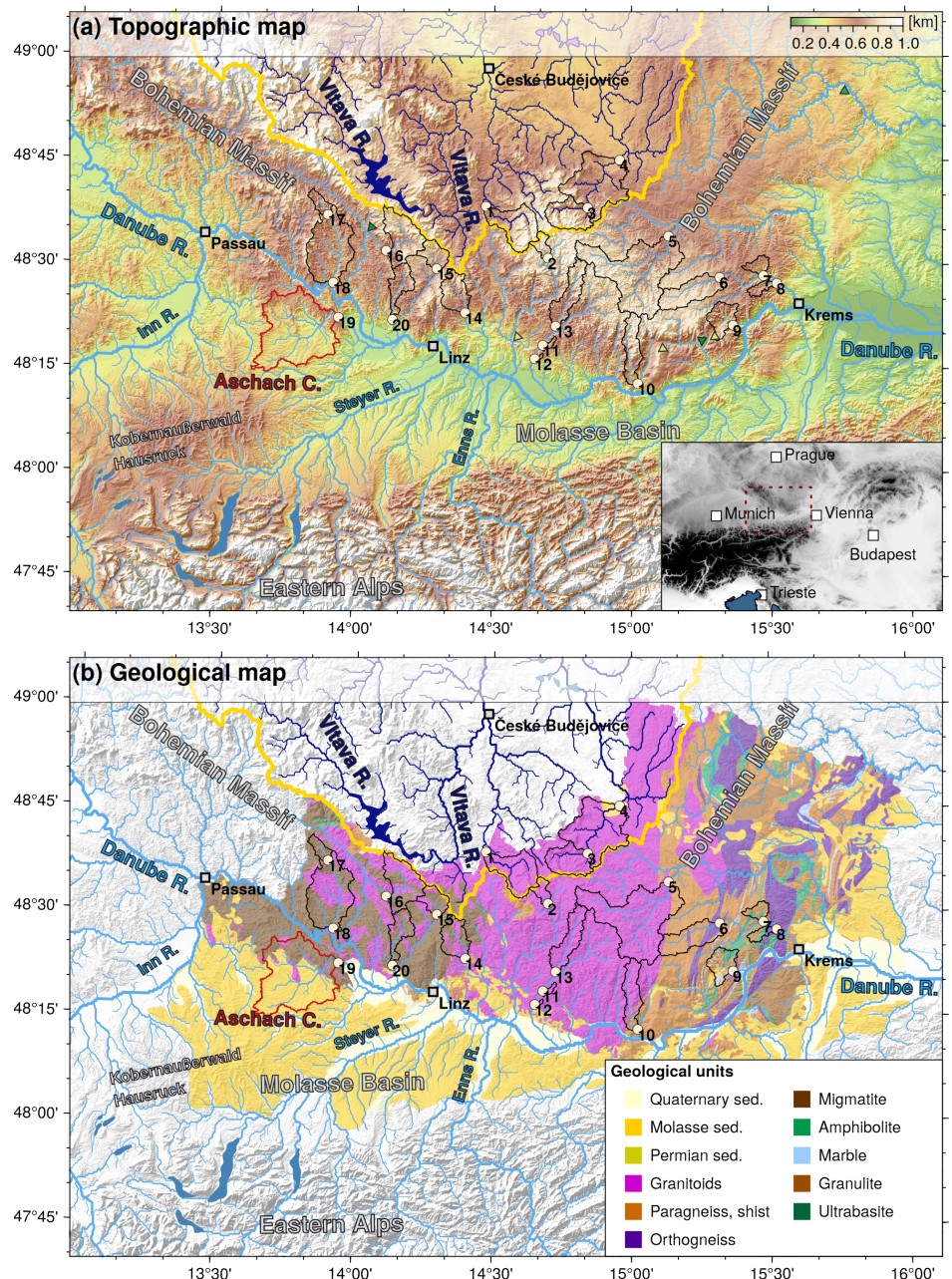

**Figure 1.** (a) Topographic map of the study site. The inset at the bottom right shows an overview map with the position of the study area indicated by the dashed red rectangle. (b) Geological map of the study area. The river networks of the Danube (light blue) and the Vltava (dark blue) are based on the global HydroRIVER dataset of HydroSHEDS (Lehner and Grill, 2013). The line thickness scales with the logarithm of the contributing drainage area. The thick yellow line indicates the continental divide between the Danube and the Vltava/Elbe drainage systems. Black polygons show the catchments upstream of the sampling locations for river sands (white circles with black outline). Sample numbers are annotated. See Tab. 1 for the catchment names. The red outline shows the position of the Aschach catchment area, with further details in Fig 9. Green and yellow triangles in (a) mark major T-shaped river junctions and prominent wind gaps in abandoned valleys. Cities (white squares with black outline) are shown for better orientation.

## 1.1 Morphological characteristics of the study region

Uplift and increasing erosion north of the Alps also had and still has far-reaching effects on the evolution of topography and the involved continental-scale drainage systems. The increase in the base level on the northern side of the Alps causes the Alps to become asymmetric on the mountain range scale with a steeper southern side and a less steep northern side. This eventually leads to the migration of the main drainage divides from the south to the north (Robl et al., 2017a). The large-scale uplift of the region also caused the reversal of original westward flow of the Paleo-Danube River (via the Rhône to the Mediterranean) to an easterly direction (via the modern Danube River to the Black Sea) (Kuhlemann and Kempf, 2002). Since flow reversal, the co-evolution and geometry of the topography in the mountainous Bohemian Massif and adjacent lowlands of the Molasse Basin are controlled by two major drainage systems that actively incise the topography that has been rising for about 8 Myr. The Danube, draining a major part of Central and Eastern Europe, flows for tens of kilometers along the boundary between the Bohemian Massif (hard crystalline rocks) and the Molasse Basin (soft sedimentary cover). In its course, lithology and, related to this, the valley morphology change repeatedly with steep flanks in places where the river has cut through crystalline basement rock to wide valley floors in the region where Molasse sediments represent the bedrock (Robl et al., 2008; Wetzlinger et al., 2023) (Figs. 1, 2). In continuation and as far-field effects of river breakthroughs in the crystalline bedrock, northeast-southwest trending ridges of soft Molasse sediments occur, which can be traced over several tens of kilometers (e.g., close to the city of Linz). Large rivers draining the northern part of the Eastern Alps (e.g., Inn River, Traun River, Enns River) bypass the outcrops of crystalline rock south of the Danube River and feed into the Danube, where the bedrock consists of soft sediments. Separated by a continental divide, the Vltava-Labe (Moldau - Elbe) river network drains the inner, and thus northern part of the bowl-shaped topography of the Bohemian Massif. Similar to the situation along the Danube and its tributaries, the area north of the continental divide is characterized by strong lithological contrasts with crystalline bedrock in the mountains and weak sediments in the central depression (e.g., České Budějovice Basin). The continental divide follows a topographic ridge over many tens of kilometers with peak elevations approaching 1.5 km and thus exhibits a relief of more than 1 km both to the north (e.g. České Budějovice: 381 m a.s.l) and to the south (e.g. Linz: 288 m a.s.l). The regions north and south of the continental divide show similar morphological characteristics. Low-amplitude and long-wavelength topography with meandering low-gradient rivers prevails in high-elevation areas and is contrasted by high amplitude and short-wavelength topography at lower elevations (Wetzlinger et al., 2023). Close to the receiving streams (i.e., the Danube River and the Vltava River south and north of the continental drainage divide, respectively) deeply incised gorges characterized by high channel gradients and morphological active hillslopes occur. T-shaped river junctions and wind gaps in abandoned valleys testify to a reorganization of the river network due to divide migration and river piracy events (Wetzlinger et al., 2023). Such a topographic pattern, where a distinct physiographic transition separates steep landforms at low elevations from gentle landforms at higher elevations, conforms in principle to relief rejuvenation due to recent uplift or base level lowering (e.g., Robl et al., 2017b). While there is a clear altitudinal dependence of the two landscape types, the elevation level of the sharp topographic change from a steep and incised to a low-gradient topography is similar in adjacent catchments but varies once the confluence points

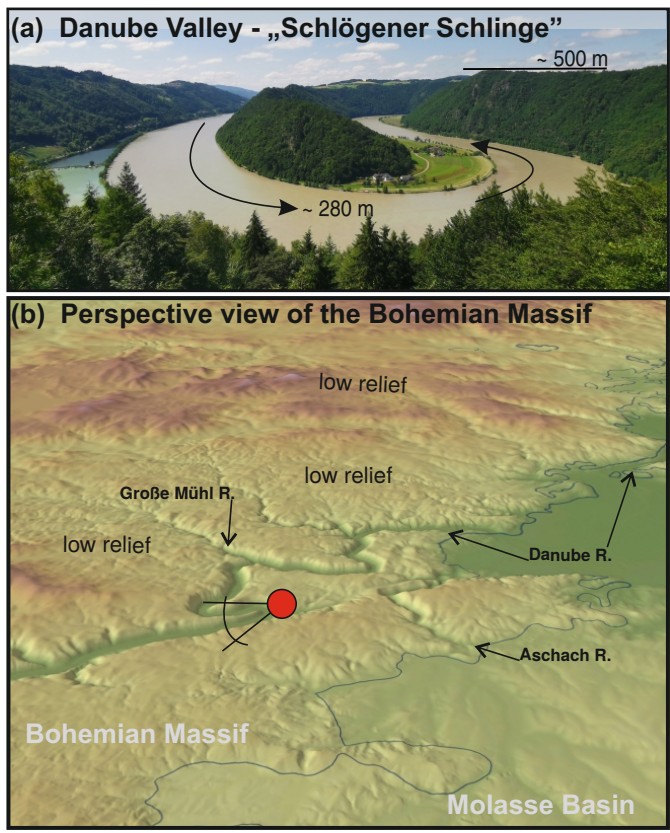

**Figure 2.** Landscape characteristics at the transition from the Molasse Basin to the Bohemian Massif. (a) Meandering Danube River at the bedrock barrier near Schlögen with the so called "Schlögener Schlinge". (b) Perspective view of the Bohemian Massif towards the north. The dark blue line marks the boundary between the soft rocks of the sedimentary cover (Molasse Basin) and the hard rocks of the crystalline basement (Bohemian Massif). The viewpoint for the Danube meander shown in (a) is marked by a red circle.

of the Danube tributaries are separated by bedrock barriers (river segments in crystalline bedrock) in the course of the Danube River.

## 1.2 Effect of lithology on eroding landscapes

The small spatial gradient in the long-term uplift pattern inferred from the almost horizontal strata of the Molasse sedimentary sequence is also confirmed by the GNSS-based vertical velocity field (Serpelloni et al., 2022, 2013). Young, large-scale uplift is generally consistent with relief rejuvenation, but cannot alone explain the difference in elevation of the physiographic transition within the Bohemian Massif. Furthermore, the topographic differences between the Bohemian Massif and the Molasse Basin cannot be attributed to this. However, there is a distinct contrast in bedrock properties at the transition between the Bohemian Massif (e.g. high-grade metamorphic and granitic rocks) and the Molasse Basin (weakly consolidated sediments) (e.g.,

Wessely, 2006). It is well recognized that spatial but also temporal variations in lithology exert a strong control over eroding landscapes (e.g., Portenga and Bierman, 2011; Scharf et al., 2013; Cyr et al., 2014; Forte et al., 2016; Gallen, 2018; Peifer et al., 2021; Haag et al., 2025). Such variations control the steepness of channels and hence catchment relief (e.g., Bernard et al., 2019), the evolution of bimodal landscapes (Anderson et al., 2023), but also the position of drainage divides and channel network topology (e.g., Zondervan et al., 2020). Furthermore, lithology has an influence on the susceptibility of steep terrain to mass movements, which in turn affects the channel morphology (e.g., Baumann et al., 2018), sediment fluxes in rivers and even bio-evolutionary pathways (Gallen, 2018; Stokes et al., 2023).

In morphological equilibrium, the influence of rock properties on the geometry of fluvially-conditioned topography is similar to that of variations in the uplift rate, whereby more resistant rocks are characterized by steeper river segments and higher catchment relief. In a transient state, however, spatial variations in substrate properties influence the pace of knickpoint migration, erosion rate and the time span to achieve a morphological equilibrium by adjusting to a new bedrock erodibility, which takes longer for erosion-resistant rocks (Forte et al., 2016; Wolpert and Forte, 2021). Since lithological contacts are typically not vertical, the outcrop of the lithological boundary migrates with progressive erosion in the river course. While a morphological equilibrium can not be established then, an erosional continuity may be approached (Perne et al., 2017). In particular, the sub-horizontal superposition of soft, easily erodible sediments on hard rocks of the crystalline basement leads to the formation of local base levels and to an increase in relief while the erosion rate decreases (e.g. Forte et al., 2016). This situation is also observed at the transition from the Molasse Basin to the Bohemian Massif.

Thus, landscapes characterized by a distinct variation in bedrock erodibility show topographic patterns that are far away from a hypothetical steady-state landscape. However, topography is still an expression of the competition between tectonically driven uplift and climatically controlled erosional surface processes. The correlation between (catchment-wide) topographic metrics and (catchment-wide) erosion rates should hold in principle, as long as the erodibility of the bedrock within the catchment does not change distinctly. This condition is largely fulfilled for the study region except for the Aschach catchment (see Fig. 1 for location).

An increase in catchment-wide erosion rate with average catchment relief and slope was already described by Ahnert (1970). Since then, the relationship between topographic metrics (e.g. channel slope, normalized channel steepness index $k_{sn}$ and hypsometry of catchments) and erosion rate has been investigated in numerous studies (e.g., Portenga and Bierman, 2011; Dixon et al., 2016), whereby a roughly linear increase in the erosion rate was observed with increased hillslope and channel steepness. This relationship breaks down in active mountain ranges, where the hillslopes have approached the critical angle (threshold slope), where excess relief is rapidly removed by landslides (e.g. Montgomery, 2001). In the latter case, erosion rates increase further with uplift rates, while topographic gradient does not (DiBiase et al., 2010; Larsen and Montgomery, 2012). However, the study region is characterized by low uplift rates and predominantly by hillslopes below the critical angle, such that the relationship between catchment-wide topographic metrics and erosion rates should hold.

## 2 Conceptual model of lithology-contrast driven relief formation

In this study, we hypothesize that relief formation and related erosion rates in the Bohemian Massif and the adjacent Molasse Basin are controlled by gradients in substrate properties. This hypothesis does not require active faults causing discontinuities in the vertical velocity field but suffices a fairly uniform uplift pattern since onset of basin inversion. Based on our conceptual model, we propose that in such a geological setting both large-scale morphological differences between the lowlands of the Molasse zone and the mountainous topography of the Bohemian Massif can arise, but also the stepped landscape in the Bohemian Massif with its pronounced physiographic transitions (Fig. 3).

At the beginning of the large-scale uplift, the bedrock was mostly covered by soft sediments of the Molasse basin (Fig. 3, Time = 0 ). Rivers form graded longitudinal profiles with uplift and erosion rates in balance. Channel gradients and the catchment relief reflect the bedrock properties of the soft Molasse sediments. With progressive erosion, rivers cut into the crystalline basement. The change in bedrock properties causes a sudden drop in erosion rate in the river segments incising the basement and thus a gradual increase in elevation (Fig. 3, Time = 1 ). As a result of these bedrock barriers, the upstream river segments that are still eroding the sedimentary cover experience a gradual increase in baselevel. This increase in baselevel results in a reduction in channel steepness, which contrasts the steepness increase in the downstream channel segment characterized by crystalline bedrock. Both river segments feature erosion rates that are lower than the uplift rate, which in turn leads to an increase in catchment relief. Besides changes in the profile geometry, the change in substrate properties can also lead to river capture events and to the reorganization of the river network (see independent river on the right). In plan view, the river makes a 90° turn and a wind gap is created. Over time a T-shaped river junction forms due to increased erosion rates of the aggressor river, which causes a gradual flow reversal of the beheaded river. Despite uniform uplift, a pronounced stepped landscape emerges with the simultaneous formation of escarpments at steep contacts between sedimentary cover and basement rocks, and low relief surfaces at various elevation levels (Fig. 3, Time = 2). The latter occur in both sedimentary and basement bedrock, which is related to the migrating boundary between the two contrasting rock types with progressive exhumation and the long response time for establishing an equilibrium channel gradient. The overlying soft sedimentary rocks on hard crystalline rocks trigger the formation of this bimodal landscape, but low relief and steep landscapes cannot be directly assigned to soft and hard bed rocks. This is because of the transient state of the topography as a result of the long response time for the adjustment of the channel gradients to the substrate properties, which change with progressive erosion.

To test the proposed conceptual model of landscape evolution for an inverted, asymmetric foreland basin, we combine field work, digital terrain analysis, cosmogenic nuclide dating and landscape evolution modeling: to determine the rates of relief formation, we compute catchment-wide erosion rates based on the concentration of cosmogenic nuclides in river sands following the pioneering approach of Granger et al. (1996). We correlate this new set of erosion rates with a variety of catchment-averaged topographic metrics to establish a link between erosion rate and topographic pattern. To investigate the timing of relief formation and the emerging topographic pattern in an area of uniform uplift, but strong differences in rock erodibility (both spatially and temporally), we apply a landform evolution model considering bedrock properties.

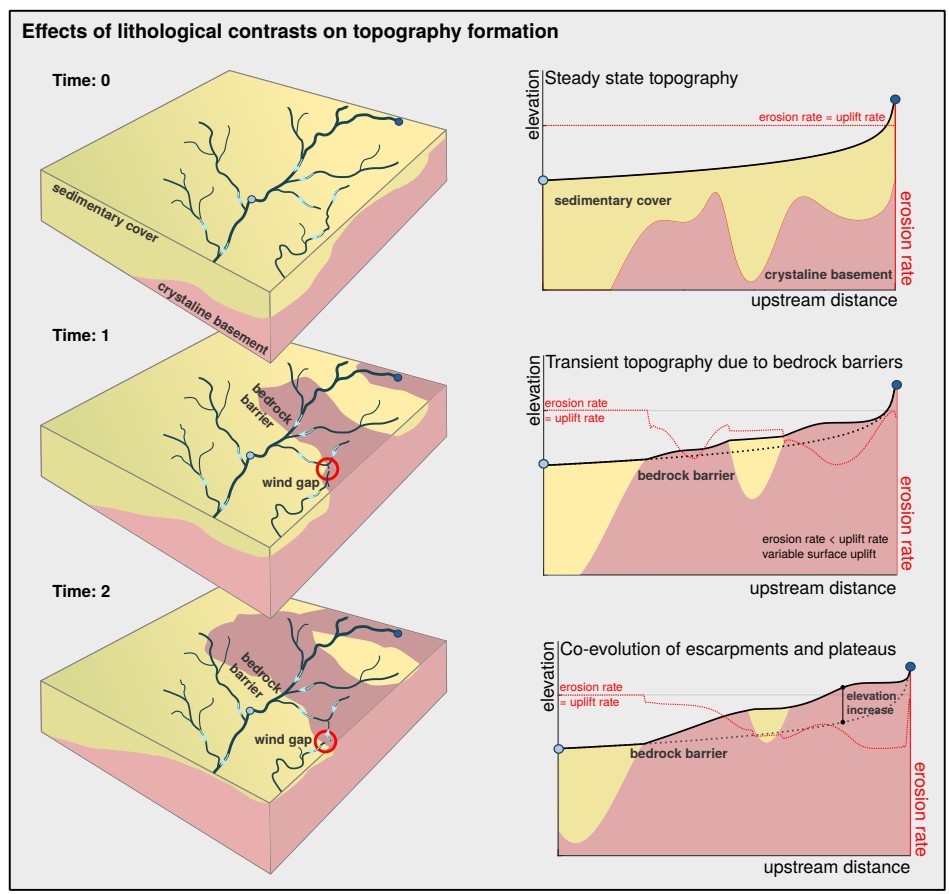

**Figure 3.** Conceptual model illustrating the evolution of topography during the uplift of an asymmetric foreland basin. The left panels show a hypothetical river network (dark blue lines) and the outcropping rocks (Neogene sediment: yellow; crystalline rocks: magenta), which change over time with uplift and ongoing erosion. Light blue arrows mark the flow direction of the streams. The red circle shows the position of the wind gap that has formed as a result of a river piracy event. The panels on the right show the longitudinal profile of the main river (solid black line). The dotted black line marks the steady-state channel profile at time = 0. Start and endpoints are indicated by the dark and light blue circles. The local erosion rate is indicated by the red dotted line. Given that this is a sketch, the distances along the trunk channel and the distribution of the rocks in the panels on the left only roughly correspond to those in the profile section on the right.

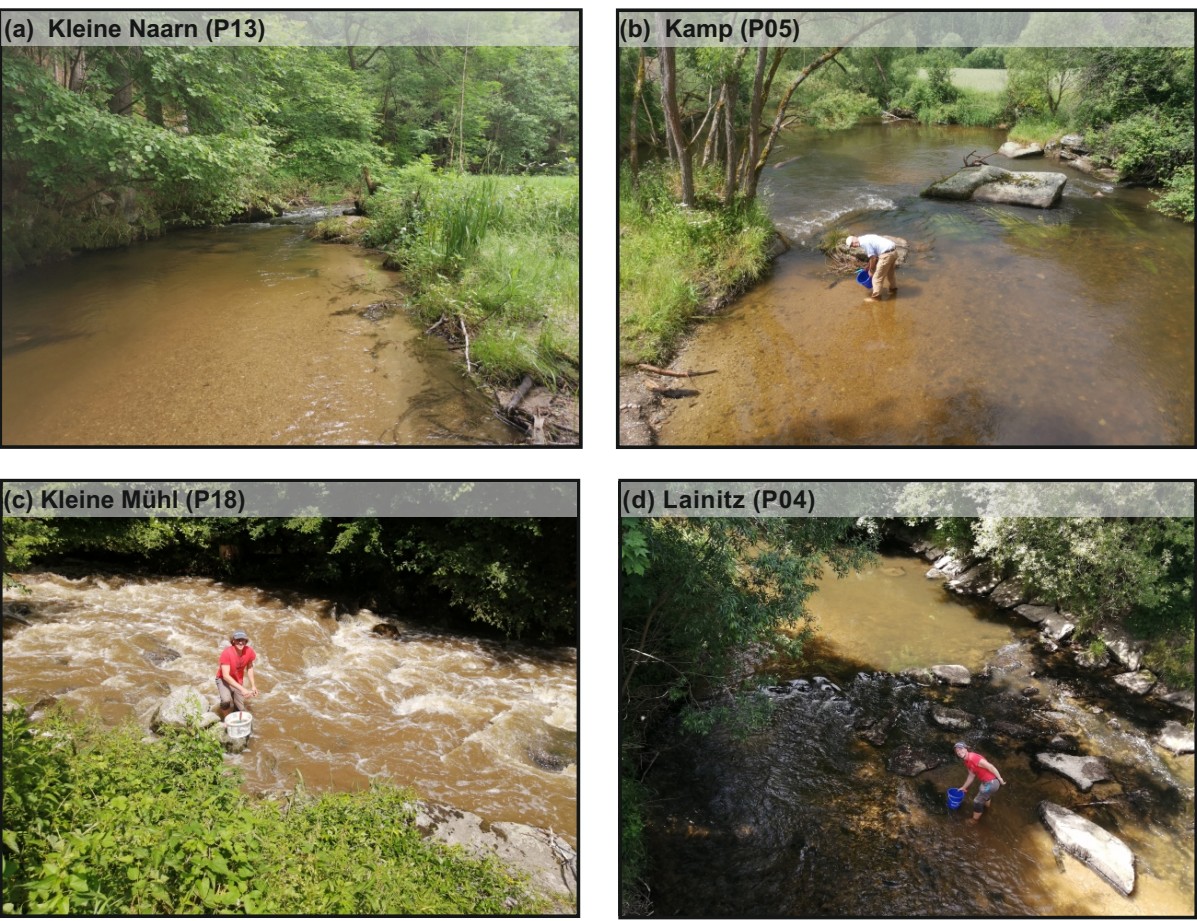

**Figure 4.** Field observations and river characteristics at sample locations. (a) Kleine Naarn River and the (b) Kamp River upstream and (c) the Kleine Mühl River downstream of the distinct physiographic transition. (d) Sampling at the upper reach of the Lainitz River, which drains into the Vltava River.

## 3  Data and Methods

High-resolution digital elevation models (DEMs) provide the basis for digital terrain analysis (drainage networks and their geometry) and for the computation of erosion rates from measured $^{10}$Be concentrations (production rates, shielding factors). We used INSPIRE DEMs for Austria (https://www.data.gv.at/) and the Czech Republic (https://geoportal.cuzk.cz/). As the two DEMs originally had different projections, both datasets were re-projected to UTM zone 33 and resampled to a spatial resolution of 10 m. The DEM tiles were merged and cropped to the study area in such a way that sampled catchments are fully represented.

## 3.1 Deriving catchment-wide erosion rates

### 3.1.1 Sampling

In a two-day field campaign (June 2021), 20 river sediment samples were taken from the active riverbeds (Figs 1, 4). Due to the predominant lithology of the Bohemian Massif, the sediments consisting of quartz-rich sands and gravels assumed to be well-mixed representatives of catchment averaged lithology and erosion rates. Sampling was conducted on both sides of the continental divide (with a focus on the Danube catchment) and sample points were selected so that both upstream and downstream of the physiographic transition were sampled. In two catchments, we applied a nested sampling strategy (Lainitz: P03, P04; Kleine Mühl: P17, P18) to detect variations in the erosion rates along the streams, in particular between the low gradient upper and the distinctly steeper lower reaches. We have also sampled several small catchments close to the Danube River that do not have large areas above the physiographic transition separating incised from low gradient landscape patches.

### 3.1.2 Preparation

All measurements of $^{10}$Be concentrations were done at the Scottish Universities Environmental Research Centre (SUERC) Accelerator Mass Spectrometry (AMS) Laboratory. For this, the samples were crushed, milled and then sieved to obtain the 250 - 500 μm fraction and magnetically separated using a roller magnetic separator. To remove carbonates and soluble oxides, the samples were first treated with warm aqua regia. Feldspar and mica minerals were removed by froth flotation (Herber, 1969). The samples were then etched three times in a 1 L solution of water, HF (40%) and $HNO_3$ (150:2:1) in a high-power ultrasonic tank to isolate the cores of the quartz grains. An aliquot of the final quartz sand was dissolved, and its purity was tested for Al, Be, Fe, Ca and Ti by ICP-OES. Pure quartz samples and process blanks (n = 3) were spiked with 0.22 mg of $^9$Be and dissolved in HF. After dissolution, the HF was evaporated and replaced by HCl. The solutions were first passed through anion exchange chromatography columns to remove Fe. The Fe-free fraction was then evaporated and the HCl was replaced by dilute $H_2SO_4$. The sulphate solutions were then passed through cation exchange chromatography columns to remove Ti, and separate Be and Al fractions. The Be fractions were precipitated as hydroxides and oxidized at $900°C$. Resulting BeO was then mixed with Nb (1:6) and pressed into copper cathodes for AMS analysis. $^{10}$Be / $^9$Be ratios were measured on the 5 MV pelletron accelerator mass spectrometry system at SUERC. All measurements were normalized to NIST SRM4325 with a nominal ratio of 2.79 x $10^{-11}$ $^{10}$Be / $^9$Be (Nishiizumi et al., 2007). The blank corrections ranged between 1.1 and 4.7% of the sample $^{10}$Be / $^9$Be ratios.

Based on $^{10}$Be concentrations of river sands, we computed catchment-wide erosion rates by employing the web-facility of the online CRONUS-Earth calculator v3.0 (Balco et al., 2008). We used 07KNSTD as our Be AMS standard and defined a sample density of $2.7\,\mathrm{g\,cm^{-3}}$. We used the latitude and longitude of the sampling location and not the catchment center, which however is of no significance due to the small catchment sizes. The mean elevation of the catchment area was calculated on the basis of the digital 10 m elevation model using standard GIS software (ArcGIS Pro). Results are given for the St, LM and LSDn scaling provided by the CRONUS-Earth calculator. For details on the different scaling models, we refer to Phillips et al.

(2016). Erosion rates were calculated without taking topographic shielding into account, as DiBiase (2018) has shown that the increased vertical attenuation length in steep topography compensates for the shielding.

## 3.2 Digital terrain analysis

Matlab and TopoToolbox (Schwanghart and Kuhn, 2010; Schwanghart and Scherler, 2014) were used for the morphometric analyses and for computing catchment statistics. The Generic Mapping Tools (GMT) were used for figure generation (Wessel et al., 2019).

To characterize the drainage system morphologically, we computed channel metrics for all streams with a catchment size $A > 0.25\,\mathrm{km}^2$. We chose the sampling point for the determination of $^{10}$Be in river sands as the outlet point for the determination of the catchment areas. This allows the measured catchment-averaged erosion rates to be correlated with the topographic features upstream of the sampling point. The gradient in flow direction (i.e., the channel slope $S = \partial H/\partial x$) was determined from the ratio of the differences in elevation ($H$) and horizontal flow length ($x$). A minimum elevation difference of $10\,\mathrm{m}$ was introduced for the computation for each river segment to achieve a certain smoothing of the inherently noisy channel slope. Based on catchment size and channel slope, we computed the normalized steepness index (Flint, 1974; Wobus et al., 2006)

$$k_{\mathrm{sn}} = A^{\theta_{\mathrm{ref}}} S. \tag{1}$$

where the reference concavity index $\theta_{\mathrm{ref}}$ was set to 0.5. This choice allows a direct comparison with the results of the morphometric analyses presented by Wetzlinger et al. (2023) for the same region and avoids the odd unit $\mathrm{m}^{0.9}$, which occurs for the widely used value $\theta_{\mathrm{ref}} = 0.45$.

To characterize the hillslope system, the distribution of elevation values for the individual catchments (e.g. mean elevation and standard deviation) and the geophysical relief ($GR$) were determined (Small and Anderson, 1998). The geophysical relief was calculated from the difference between the maximum elevation $H_{\mathrm{max}}$ within a moving window with radius $250\,\mathrm{m}$ and the respective elevation of the window center point ($H$).

$$GR = H_{\mathrm{max}} - H \tag{2}$$

The window size was chosen so that valley cross-sections with ridges and valley floors are well represented and are not yet obscured by the general increase in height in the direction of the main ridge.

We computed channel slope, normalized steepness index and geophysical relief for all investigated catchments and determined catchment-wide properties such as mean, median and standard deviation of the property (Fig. 5a). We applied elevation masks to compute the topographic properties for different elevation slices. Q1 represents the lowest elevation quarter (lowest 25% of the catchment) and hence regions near the base level. The interquartile range addresses the central 50% of the elevation range of the catchment (Q2 + Q3). Elevation quarter Q4 covers the highest 25% of the catchment and hence is representative for the region near the principal divide (Fig. 5b). Additionally, we determined the area fraction of topographic metric that exceeds or falls below certain thresholds (Fig. 5c). This was done for entire catchments but also for the three different elevation slices. Since the signal of morphological response (i.e. channel slope, geophysical relief) to temporal variation in uplift rate

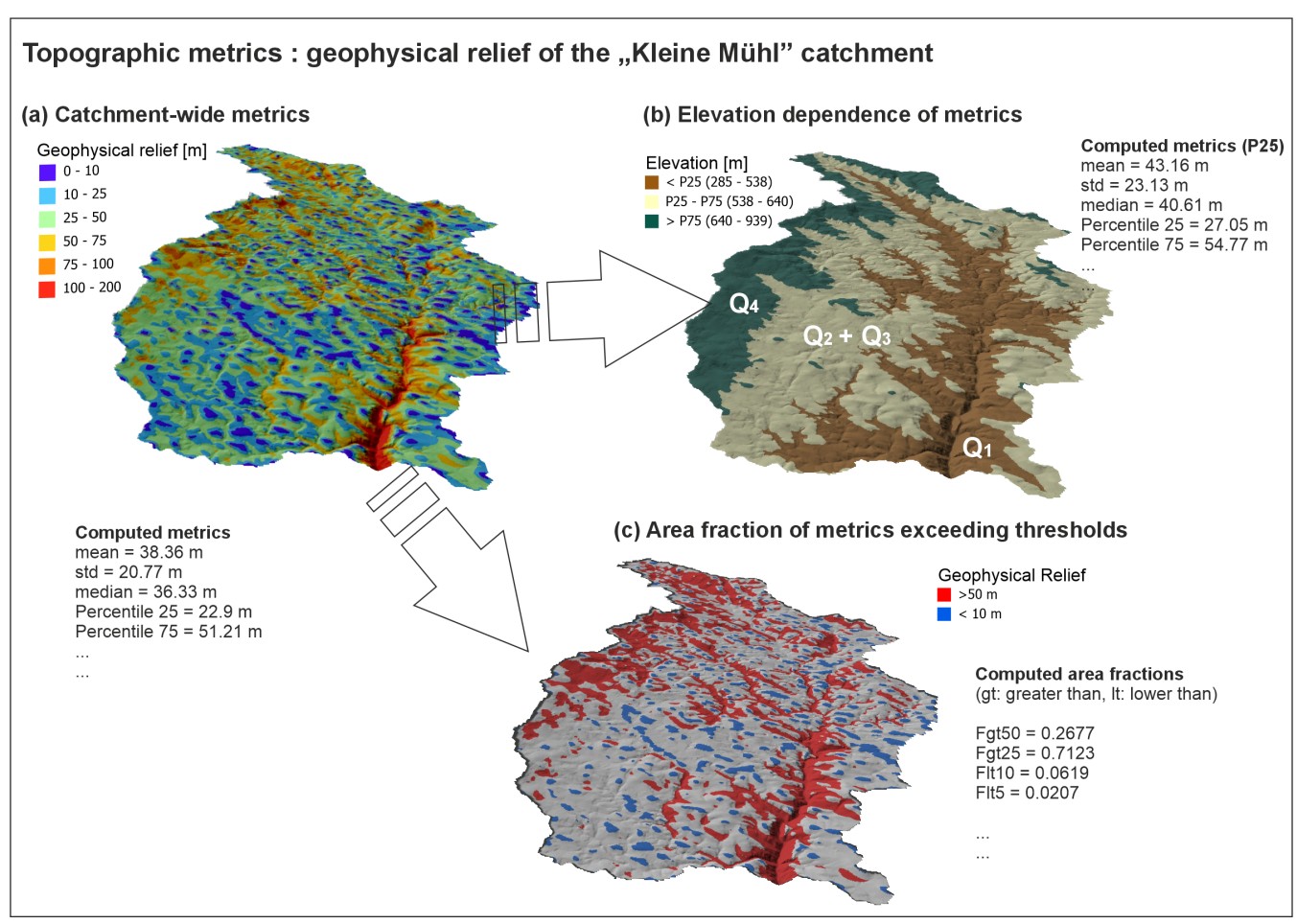

**Figure 5.** Characterizing catchment topography using the example of the geophysical relief of the Kleine Mühl catchment. (a) Catchment-wide metrics provide mean, median, standard deviation and percentiles of the investigated metric. (b) Using elevation masks to compute the properties of (a) for the lowest (Q1) and highest (Q4) elevation quarter and for the interquartile range (Q2 + Q3). (c) Determination of the area fraction where a threshold value for a topographic metric is exceeded or undercut. Threshold values in the example are geophysical relief > 50 m (red) and < 10 m (blue).

(but also in bedrock erodibility and in climate) in fluvial landscapes propagates upstream from a stable base level, the segmentation of transient catchments into elevation slices facilitates its detection. The morphological adjustment is also reflected in a spatially variable (altitude-dependent) erosion rate and, in attenuated form, in a changed catchment-wide erosion rate. Hence, we correlate topographic metrics for the entire catchment and for the three elevation slices with the catchment-wide erosion rates.

To determine the degree of correlation, we computed the Pearson linear correlation coefficient between the catchment-wide erosion rates of the analyzed catchments and their topographic metrics. This was done for catchment-wide topographic metrics, but also for different elevation levels within each catchment and for the proportion of area where topographic metrics are greater or less than a specified threshold (Fig. 5).

### 3.3    Modeling the topographic evolution

To describe the time-dependent evolution of the principal topographic features of the study region, we employ OpenLEM and its shared stream power approach (Hergarten, 2020). In this model, the change in surface elevation $H$ with time $t$ is described by

$$\frac{\partial H}{\partial t} = U - E \tag{3}$$

where $U$ and $E$ is uplift and erosion rate, respectively. In the shared stream power model, rivers can be in a state between detachment- and transport-limited conditions and erosion rate $E$ depends on the local stream power $A^m S^n$ and the sediment flux $Q$:

$$\frac{E}{K_d} + \frac{Q}{K_t A} = A^m S^n \tag{4}$$

We use the simplest choice for the exponents, m = 0.5 and n = 1. While the bedrock erodibility $K_d$ controls the detachment of material from the riverbed, $K_t$ describes the ability to transport sediment. In the spirit of the model, the two processes of bedrock detachment and sediment transport share the local stream power of the river. As discussed by Hergarten (2021), the specific case of spatially uniform erosion ($Q = EA$) can be described by an effective erodibility $K$ defined by the relation

$$\frac{1}{K} = \frac{1}{K_d} + \frac{1}{K_t} \tag{5}$$

Beside fluvial erosion, the model can effectively handle sediment transport and deposition whenever the transport capacity is exceeded. The three-dimensional geometry of the geological units (i.e., basement of the Bohemian Massif, sediments of the Neogene Molasse basin and alluvial deposits, which form during the simulation) is represented by the OpenLEM layer approach (Hergarten, 2024). Layers in OpenLEM allow defining the spatial position of the respective unit and enable the assignment of bedrock-specific erodibilities. By defining large $K_d$ values for easily erodible bedrock (e.g. alluvial sediments), rivers approach a transport-limited state.

The model setup includes three material layers ($L_2$ - $L_0$ from bottom to top) that define the spatial distribution of three different rock types of the study region (Kröll et al., 2006, 2001; Kröll and Wessely, 2001). The bottom layer $L_2$ describes

the basement rock. As discussed by Hergarten (2020), the findings of Guerit et al. (2019) tentatively suggest $K_d/K_t \approx 1.6$. We therefore assume the values $K_d = 2.6 \ \mathrm{Myr}^{-1}$ and $K_d = 1.625 \ \mathrm{Myr}^{-1}$. According to Eq. 5 these values correspond to an effective erodibility of $K = 1 \ \mathrm{Myr}^{-1}$, which is suitable for granitic rock (Fox et al., 2014).

This layer follows to the complex geometry of the topography of the Alps and the Bohemian Massif in the south and north respectively, and the subsurface geometry of the Molasse basin in between. To achieve this, the contour lines of the sediment thickness of the Molasse Basin (Kröll et al., 2006; Kröll and Wessely, 2001) and a smoothed digital elevation model of the surrounding mountains were amalgamated to form a common bedrock surface that represents the region without the sedimentary infill of the Molasse Basin but also without valleys and interfluves. In order to compensate for the uplift in the course of basin inversion, this bedrock surface was lowered by 800 m. Uplift of 800 m over the last 10 Myr represents a minimum assumption because elevations within the Molasse Basin in the Kobernaußerwald - Hausruck region come close to this value (Baumann et al., 2018).

The layer $L_1$ represents the Neogene sediments of the Molasse basin and superimposes $L_2$. To define this layer, we fill the depression that occurs in $L_2$ between the Alps and the Bohemian Massif to sea level. This results in a peneplain in the realm of the Molasse Basin at sea level. Assuming that $K_d$ is 10 times as large as for $L_2$ ($K_d = 26 \ \mathrm{Myr}^{-1}$) and keeping $K_t = 1.625 \ \mathrm{Myr}^{-1}$ allows the river to effectively erode its bed and rapidly approach its transport capacity. The uppermost layer $L_0$ describes unconsolidated deposits ($K_d = \infty, K_t = 1.625 \ \mathrm{Myr}^{-1}$). This parameter choice enforces a transport-limited river once it flows over its own deposits. The thickness of this layer is zero at onset of the simulation. We defined a uniform uplift rate of $100 \ \mathrm{m \, Myr}^{-1}$ for the entire region, with the boundaries remaining at a constant base level. This is consistent with the knowledge that uplift and basin inversion of the region started about 8 Myr ago (Gusterhuber et al., 2012). To achieve a principal eastward flow direction, the entire initial topography was tilted so that the western boundary is at 20 m and drops linearly to 0 m at the eastern boundary. With a west-east extent of 340 km, this results in a topographic gradient of $6 \times 10^{-5}$.

The influence of a large river such as the Danube is considered by placing the origin of a river with about 3 times the catchment size of the model domain at the center of the western model boundary. Since OpenLEM cannot take into account rivers with a given discharge and sediment flux entering the domain, the extension for variable precipitation was used. In addition to the computation of orographic precipitation (Hergarten and Robl, 2022) it allows freely definable precipitation patterns. A river entering the domain is defined by increasing the precipitation at the respective boundary cell by the hypothetic catchment size of the river. Since this cell reaches a steady state rapidly, the incoming sediment flux can be defined by adjusting the uplift rate at the respective cell. To avoid excessive incision, an equilibrium sediment load is assigned at the source point that corresponds to the catchment size and the uplift rate in the model domain.

**Table 1.** [10]Be concentrations and derived erosion rates from 20 catchments draining the Bohemian Massif. Isotope ratios were normalized to NIST SRM 4325 using $^{10}\text{Be}/^{9}\text{Be} = 2.79 \times 10^{-11}$. Blank corrections (Blk corr.) for [10]Be concentrations are < 2%, except for sample P07 where the blank correction is 4.75%.

| Sample | Name | Catchment characteristics | | | | | Cosmogenic nuclides in river sands | | | |
| --- | --- | --- | --- | --- | --- | --- | --- | --- | --- | --- |
| | | Lat [°N] | Long [°E] | $E_{mean}$ [m] | A [km$^2$] | Quartz [g] | $^{9}$Be spike [µg] | Blk corr. | $^{10}$Be [atoms g$^{-1}$] | $^{10}$Be / $^{9}$Be |
| P01 | Maltsch | 48.6439 | 14.4728 | 804 | 107.7 | 22.087 | 222.7 ± 3.1 | 1.2% | 220773 ± 6292 | 3.318E-13 ± 8.055E-15 |
| P02 | Flammbach | 48.5206 | 14.6981 | 911 | 20.9 | 23.205 | 221.2 ± 3.1 | 1.3% | 196156 ± 5050 | 3.121E-13 ± 6.563E-15 |
| P03 | Lainsitz | 48.6416 | 14.8413 | 835 | 60.8 | 20.866 | 222.4 ± 3.1 | 1.8% | 133297 ± 3493 | 1.907E-13 ± 4.064E-15 |
| P04 | Lainsitz | 48.7563 | 14.9569 | 638 | 268.3 | 22.476 | 223.8 ± 3.2 | 1.4% | 159225 ± 4623 | 2.428E-13 ± 6.019E-15 |
| P05 | Kamp | 48.5733 | 15.1335 | 800 | 304.3 | 23.158 | 223.5 ± 3.2 | 1.4% | 156407 ± 4702 | 2.461E-13 ± 6.388E-15 |
| P06 | Große Krems | 48.4740 | 15.3182 | 806 | 91.2 | 9.287 | 224.2 ± 3.2 | 1.9% | 242455 ± 8492 | 1.531E-13 ± 4.772E-15 |
| P07 | Gföhler Bach | 48.4776 | 15.4762 | 549 | 21.4 | 6.801 | 224.7 ± 3.2 | 4.7% | 126196 ± 4519 | 5.998E-14 ± 1.794E-15 |
| P08 | Reichaubach | 48.4592 | 15.5241 | 561 | 17.4 | 22.649 | 223.2 ± 3.2 | 1.8% | 127792 ± 6222 | 1.976E-13 ± 9.011E-15 |
| P09 | Spitzer Bach | 48.3590 | 15.3657 | 617 | 51.6 | 16.133 | 224.4 ± 3.2 | 1.7% | 154530 ± 5268 | 1.691E-13 ± 5.115E-15 |
| P10 | Kleine Ysper | 48.2190 | 15.0223 | 730 | 67.8 | 22.884 | 222.6 ± 3.1 | 1.8% | 122068 ± 4644 | 1.913E-13 ± 6.592E-15 |
| P11 | Hiasbach Naarn | 48.3108 | 14.6802 | 554 | 5.3 | 23.260 | 222.7 ± 3.1 | 1.7% | 126756 ± 3704 | 2.017E-13 ± 5.011E-15 |
| P12 | Naarn tributary | 48.2781 | 14.6517 | 468 | 1.9 | 22.584 | 222.1 ± 3.1 | 1.6% | 141965 ± 4014 | 2.196E-13 ± 5.235E-15 |
| P13 | Kleine Naarn | 48.3570 | 14.7260 | 751 | 74.9 | 21.185 | 222.4 ± 3.1 | 1.9% | 124767 ± 3574 | 1.814E-13 ± 4.370E-15 |
| P14 | Große Gusen | 48.3877 | 14.3997 | 715 | 63.8 | 24.495 | 222.1 ± 3.1 | 1.3% | 167103 ± 4594 | 2.793E-13 ± 6.451E-15 |
| P15 | Rodl | 48.4934 | 14.2959 | 783 | 40.2 | 23.129 | 224.5 ± 3.2 | 0.8% | 217425 ± 5228 | 3.381E-13 ± 6.473E-15 |
| P16 | Steinerne Mühl | 48.5359 | 14.1127 | 780 | 105.8 | 22.902 | 223.5 ± 3.2 | 1.1% | 164387 ± 4813 | 2.549E-13 ± 6.421E-15 |
| P17 | Kleine Mühl | 48.6206 | 13.9003 | 638 | 22.5 | 21.432 | 224.1 ± 3.2 | 1.1% | 172518 ± 5467 | 2.498E-13 ± 6.967E-15 |
| P18 | Kleine Mühl | 48.4574 | 13.9222 | 592 | 200.2 | 21.454 | 223.8 ± 3.2 | 1.6% | 125496 ± 3525 | 1.829E-13 ± 4.317E-15 |
| P19 | Aschach | 48.3742 | 13.9425 | 435 | 323.6 | 19.403 | 224.1 ± 3.2 | 1.7% | 130376 ± 3842 | 1.717E-13 ± 4.316E-15 |
| P20 | Kleine Rodl | 48.3700 | 14.1383 | 636 | 51.4 | 29.000 | 223.9 ± 3.2 | 1.5% | 95802 ± 2352 | 1.886E-13 ± 3.666E-15 |

**Table 2.** Continued Table 1

| Sample | ST Erosion Rate [g/cm²/yr] | [m Myr⁻¹] | ST Uncertainty Internal [m Myr⁻¹] | External [m Myr⁻¹] | LM Erosion Rate [g/cm²/yr] | [m Myr⁻¹] | LM Uncertainty Internal [m Myr⁻¹] | External [m Myr⁻¹] | LSDn Erosion Rate [g/cm²/yr] | [m Myr⁻¹] | LSDn Uncertainty Internal [m Myr⁻¹] | External [m Myr⁻¹] |
|---|---|---|---|---|---|---|---|---|---|---|---|---|
| P01 | 0.00667 | 24.7 | 0.713 | 2.10 | 0.00685 | 25.4 | 0.732 | 2.06 | 0.00678 | 25.1 | 0.724 | 1.66 |
| P02 | 0.00820 | 30.4 | 0.789 | 2.55 | 0.00841 | 31.1 | 0.810 | 2.49 | 0.00834 | 30.9 | 0.802 | 2.01 |
| P03 | 0.01150 | 42.7 | 1.130 | 3.58 | 0.01180 | 43.6 | 1.150 | 3.49 | 0.01160 | 43.1 | 1.140 | 2.80 |
| P04 | 0.00822 | 30.4 | 0.893 | 2.59 | 0.00841 | 31.2 | 0.913 | 2.53 | 0.00829 | 30.7 | 0.900 | 2.04 |
| P05 | 0.00949 | 35.2 | 1.070 | 3.00 | 0.00971 | 36.0 | 1.090 | 2.93 | 0.00959 | 35.5 | 1.080 | 2.37 |
| P06 | 0.00604 | 22.4 | 0.794 | 1.96 | 0.00622 | 23.0 | 0.817 | 1.93 | 0.00615 | 22.8 | 0.808 | 1.58 |
| P07 | 0.00970 | 35.9 | 1.300 | 3.14 | 0.00992 | 36.7 | 1.330 | 3.08 | 0.00973 | 36.0 | 1.300 | 2.51 |
| P08 | 0.00966 | 35.8 | 1.760 | 3.35 | 0.00988 | 36.6 | 1.800 | 3.30 | 0.00970 | 35.9 | 1.760 | 2.77 |
| P09 | 0.00830 | 30.7 | 1.060 | 2.67 | 0.00851 | 31.5 | 1.080 | 2.62 | 0.00836 | 31.0 | 1.070 | 2.13 |
| P10 | 0.01160 | 42.9 | 1.640 | 3.78 | 0.01180 | 43.8 | 1.680 | 3.71 | 0.01160 | 43.1 | 1.650 | 3.05 |
| P11 | 0.00967 | 35.8 | 1.050 | 3.04 | 0.00989 | 36.6 | 1.080 | 2.97 | 0.00970 | 35.9 | 1.060 | 2.38 |
| P12 | 0.00802 | 29.7 | 0.849 | 2.52 | 0.00822 | 30.5 | 0.869 | 2.46 | 0.00805 | 29.8 | 0.852 | 1.97 |
| P13 | 0.01150 | 42.7 | 1.230 | 3.61 | 0.01180 | 43.6 | 1.260 | 3.53 | 0.01160 | 43.0 | 1.240 | 2.84 |
| P14 | 0.00828 | 30.7 | 0.851 | 2.59 | 0.00849 | 31.4 | 0.872 | 2.53 | 0.00836 | 31.0 | 0.859 | 2.03 |
| P15 | 0.00665 | 24.6 | 0.599 | 2.06 | 0.00684 | 25.3 | 0.616 | 2.02 | 0.00675 | 25.0 | 0.608 | 1.61 |
| P16 | 0.00888 | 32.9 | 0.971 | 2.79 | 0.00909 | 33.7 | 0.994 | 2.73 | 0.00897 | 33.2 | 0.982 | 2.21 |
| P17 | 0.00755 | 28.0 | 0.895 | 2.41 | 0.00774 | 28.7 | 0.918 | 2.36 | 0.00762 | 28.2 | 0.903 | 1.91 |
| P18 | 0.01010 | 37.4 | 1.060 | 3.16 | 0.01030 | 38.2 | 1.080 | 3.09 | 0.01010 | 37.5 | 1.060 | 2.47 |
| P19 | 0.00854 | 31.6 | 0.941 | 2.69 | 0.00874 | 32.4 | 0.963 | 2.63 | 0.00857 | 31.7 | 0.944 | 2.11 |
| P20 | 0.01380 | 51.2 | 1.260 | 4.26 | 0.01410 | 52.2 | 1.290 | 4.14 | 0.01380 | 51.3 | 1.270 | 3.30 |

## 4 Results and Discussion

We examined 20 catchments in the South Bohemian Massif, of which 17 are located south of the continental divide and drain into the Danube (Fig. 1). One of these catchments is located on the orographic right-hand side of the Danube River (catchment 19: Aschach) and extends far into the Neogene Molasse Basin (Fig. 1). Three catchments are located north of the continental divide and are tributaries of the Vltava River. A detailed morphometric analysis for the southern Bohemian Massif has already been carried out by (Wetzlinger et al., 2023) and the key findings are presented in the Sect. 1.1 accompanied by figures 1, 2 and 4. For a study region-wide overview of the spatial distribution of low relief surfaces and incised landscapes, please refer to figure 1 of the supplemental Zenodo repository (Robl, 2025) showing geophysical relief and the drainage network color-coded for $k_{sn}$. The supplement also contains detailed morphometric analyses for each of the 20 catchments and a comprehensive table with catchment-wide topographic metrics as shown in Figs 6 and 7.

In this study, we focus on catchment-wide topographic metrics that characterize both the hillslope and the drainage system and allow a direct comparison with catchment-averaged erosion rates. For reproducibility, we apply the area fraction of the catchments with very low or high relief (hillslope system) as well as the fraction of channels with large or small channel steepness (drainage system), instead of an expert-based mapping of the two different landscape types (i.e., low relief and incised landscapes). Since the erosional signal migrates from a (local) base level upstream, we subdivide the catchment areas into elevation quarters to identify changes in topography with elevation and their effect on the measured erosion rates.

### 4.1 Erosion rates expressed by catchment-wide topographic metrics

Here we present the first catchment-wide erosion rates of the Southern Bohemian Massif and their correlation with catchment-averaged topographic metrics derived from the geophysical relief, channel slope and normalized channel steepness. Our results show significant links between landscape geometry and the rate of topographic adjustment (Figs. 6 - 8). Catchment-wide erosion rates of the Southern Bohemian Massif are low with 22 to 51 $\mathrm{m\,Myr^{-1}}$ but differ by a factor of more than 2 (Fig. 6a, Tab. 1). Differences in the individual scaling methods (St, Lm, LSDn) of the CRONUS-Earth calculator v 3.0 (Balco et al., 2008) for converting $^{10}$Be concentrations into erosion rates are less than 1 $\mathrm{m\,Myr^{-1}}$ or less than 3% and smaller than the computed external uncertainties of the method. All catchments predominantly consist of quartz-rich rocks (granites and gneisses). However, amphibolites and marbles (quartz-free) occur to a lesser extent in the eastern catchments (Fig. 1, catchments 7, 8, 9).

Highest erosion rates occur at fairly north-south draining catchments of Danube tributaries with a large length-to-width ratio (i.e. catchments 10: Kleine Ysper, 13: Kleine Naarn, 20: Kleine Rodl). The lowest erosion rates occur in rivers that feature a large increase in catchment size with flow length and have a significant vertical and horizontal distance to the active receiving stream. This applies to catchments on both sides of the continental divide (i.e. 1: Maltsch; 6: Große Krems; 15: Rodl), where the Danube and the Vltava represent the base level south and north of the continental divide, respectively. However, in the catchments with erosion rates at the lower end of the observed range, the trends are inconclusive and there are some outliers (i.e. 17: Kleine Mühl, upper reach; 3: Lainsitz upper reach).

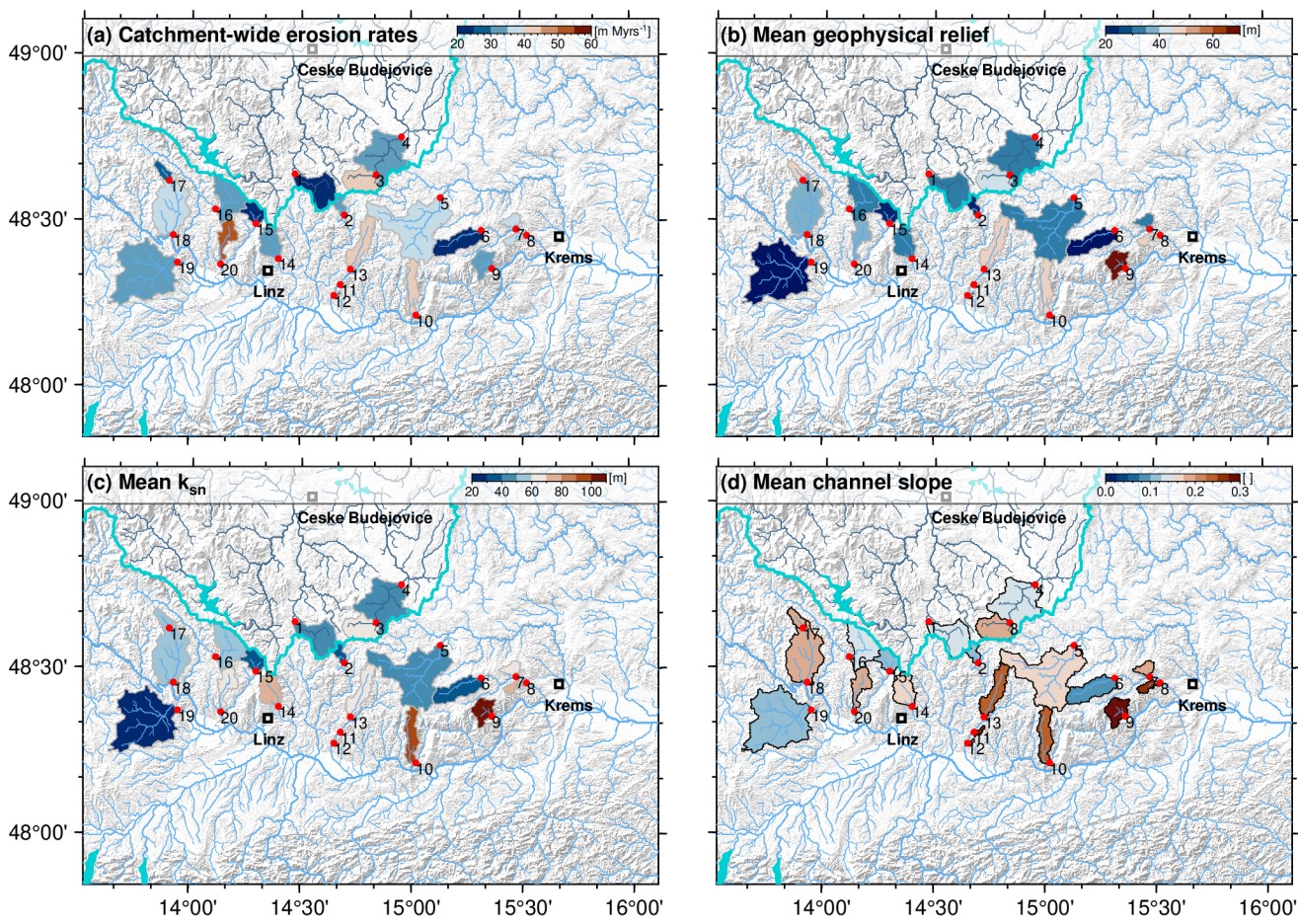

**Figure 6.** Maps showing (a) catchment-wide erosion rates in comparison with (b) mean geophysical relief, (c) mean $k_{sn}$ and (d) mean channel slope. The river networks of the Danube (light blue) and the Vltava (dark blue) are based on the global HydroRIVER dataset of HydroSHEDS (Lehner and Grill, 2013). The thick cyan line indicates the continental divide between the Danube and the Vltava/Elbe drainage systems. Polygons show the catchments upstream of the sampling locations for river sands (red circles). Sample numbers are annotated (see Tab. 1).

It is evident that the measured erosion rates are related to catchment-wide topographic metrics (compare Fig. 6a with Fig. 6 b-d). In general, an increase in catchment-wide erosion rates is also associated with an increase in mean geophysical relief, channel steepness ($k_{sn}$) and channel slope ($S$). However, anomalies also occur here. For example, catchment 9 (Spitzer Bach) has a low average erosion rate of $30 \, \mathrm{m} \, \mathrm{Myr}^{-1}$, although it has the highest values of geophysical relief, $k_{sn}$, and channel slope of all the catchments studied. Catchment 20 (Kleine Rodl) shows high erosion rates exceeding $50 \, \mathrm{m} \, \mathrm{Myr}^{-1}$ and above average values in $k_{sn}$ and channel slope but only moderate values of geophysical relief. The average geophysical relief, $k_{sn}$ and channel slope in catchment 19 (Aschach) are at the lower limit of the observed value ranges in the study region, but the catchment-wide erosion rate of almost $32 \, \mathrm{m} \, \mathrm{Myr}^{-1}$ is in the range of catchments with a topography that is significantly more incised by rivers (greater relief) and also steeper. However, the Aschach catchment is the only investigated catchment that lies predominantly in the Molasse zone, and the bedrock consists mainly of Neogene sediments (Fig. 1b). The crystalline basement rocks of the Bohemian Massif are only exposed near the Danube valley. There, channel slope, $k_{sn}$ and geophysical relief are significantly higher than in the rest of the catchment area where Molasse sediments are exposed.

To demonstrate how closely individual catchments follow the given correlation or deviate from the regression line, we show three scatter plots with topographic metric against erosion rate for relief, channel steepness and $k_{sn}$ (Fig. 7). It turns out that many of the statistical properties of the three topographic measures examined show a moderate ($0.5 \leq |r| < 0.7$) to high ($0.7 \leq |r| < 0.9$) correlation with the determined erosion rate and $P$ values below 5%. The majority of catchments seem to follow a distinct relationship between erosion rate and selected topographic metric, although with a large scatter around the regression line. Two of the investigated catchments deviate particularly strongly from the linear correlation between erosion rate and topographic metrics and form outliers (Fig. 7, red dots). Assuming a linear relationship between erosion rate and topographic metric, catchment 20 (Kleine Rodl) has erosion rates that are "too high" and catchment 9 (Spitzerbach) has erosion rates that are "too low" for their topographic properties. If these two catchments are not taken into account, the degree of correlation increases distinctly for most erosion rate - topographic metric couples (Fig. 7, 8).

Taking all 20 catchments into account, the correlation between the erosion rate and mean geophysical relief, mean channel slope and mean $k_{sn}$ is low to moderate (Fig. 6 b-d, 7 a-c). Without the two outlier catchments, however, the degree of correlation is high. However, there are a number of additional landscape metrics that are more strongly correlated with erosion rate or provide additional insights into the erosion dynamics. Thus, the area fraction of geophysical relief < 10 m and channel slope < 0.1 shows a negative correlation with erosion rate, indicating that erosion rate decreases with the increasing proportion of low-gradient hillslopes and channels and increases with the proportion of steep landforms (Fig. 7 d, e). Interestingly, catchment-wide erosion rate shows the highest degree of correlation with the standard deviation of the three topographic metrics examined. This applies to metrics that consider the entire catchment area, but also to those that refer to the lowest elevation quarter (Q1) and the interquartile range in elevation (Q2+Q3) (Fig. 7 f, h). The highest 10 percent of all values, as shown in the example "Percentile 90 $k_{sn}$ in Q1" (Fig. 7i), show a significantly higher positive correlation with the erosion rate than the mean values of these metrics. The correlation between erosion rate and catchment relief (Fig. 7g), a measure already associated with erosion rate by Ahnert (1970) is significantly weaker than all the other metrics analyzed.

For individual catchments, clear relationships between the geometry of hillslopes and channels and the rate of erosion become evident. Alongside the lowest erosion rate, catchment 6 (Große Krems) shows catchment relief at the lower end of the investigated catchments, the lowest mean geophysical relief and the largest proportion of area with channel slope < 0.1 and the second largest area fraction of geophysical relief < 10 m. Only catchment 19 (Aschach), which is mainly located in the Molasse Basin, has a larger proportion of low relief surfaces. However, the catchment is also characterized by the lowest

value in the mean channel slope and as well as by low $k_{sn}$ values. At the upper end of the measured erosion rates, catchment 10 (Kleine Ysper) shows the highest catchment relief, and also increased values in the other metrics of the geophysical relief, channel slope and $k_{sn}$. Among others, the area fraction with geophysical relief > 25 m is at the upper end and the area fraction with channel slope < 0.1 (i.e. low gradient channels) and geophysical relief < 10 m is at the lower end of the measured values. Even the two outlier catchments follow the relationship between erosion rate and $k_{sn}$, specifically that catchments with large

standard deviation in $k_{sn}$ and with large values in the 90[th] percentile in $k_{sn}$ also have large erosion rates and vice versa.

The correlations shown in the scatter plots and discussed above exemplify the degree of correlation between erosion rate and relief representative of the hillslope system and channel slope, respectively channel steepness representing the drainage system. The presentation of all Pearson's linear correlation coefficients provides a systematic overview of which topographic metrics in the study area show the highest degree of correlation with catchment-wide erosion rates (Fig. 8). Apparently, many

of the metrics shown for geophysical relief, channel slope and $k_{sn}$ are not independent of each other.

Considering all 20 catchments, geophysical relief shows the lowest, but still significant, correlation with the erosion rate of the three analyzed topographic metrics. If the entire catchment area is considered, the correlation coefficient for the mean, standard deviation (std), median, 10[th] percentile (P10), 25[th] percentile (P25), 75[th] percentile (P75), and 90[th] percentile (P90) of the geophysical relief is between 0.4 and 0.5. The degree of correlation of the area fraction with a geophysical relief greater

than 25 m (Fgt25) is moderate with $r > 0.5$. The correlation between the erosion rate and the area fraction of geophysical relief smaller 10 and 5 meters is moderately negative and in line with the above-described increase in the erosion rate with relief.

Larger differences in the degree of correlation are found when elevation levels are examined separately. For the lowest elevation quarter (Q1) and for the interquartile range of elevation (Q2+Q3), std and Fgt25 show a moderate correlation with erosion rate. Although the highest elevation quarter (Q4) shows a distribution of correlation coefficients similar to that of the

385 entire catchment, the correlations are generally lower. Without considering the two outlier catchments, the degree of correlation increases significantly and consistently reaches values of $r > 0.5$ and for some metrics $r > 0.7$. However, in the highest elevation quarter, the already low degree of correlation drops slightly for all metrics.

Channel slope shows a moderate correlation with erosion rate, whereby most of the metrics have very similar correlation coefficients. Excluding the two outlier catchments, several metrics with $r > 0.7$ correlate distinctly with erosion rate. The

390 strongest positive correlations are between standard deviation in channel slope, 75[th] percentile in channel slope, and area fraction with channel slopes larger 30%. This is found for the entire area of the catchment but even stronger for the regions defined by the lower elevation quarter (Q1) and the interquartile range in elevation (Q2+Q3). There is also a significant negative correlation between the erosion rate and the area fraction with slope less than 10% or less than 5%. Like the geophysical relief, the degree of correlation for the channel slope also decreases significantly in the highest elevation quarter (Q4).

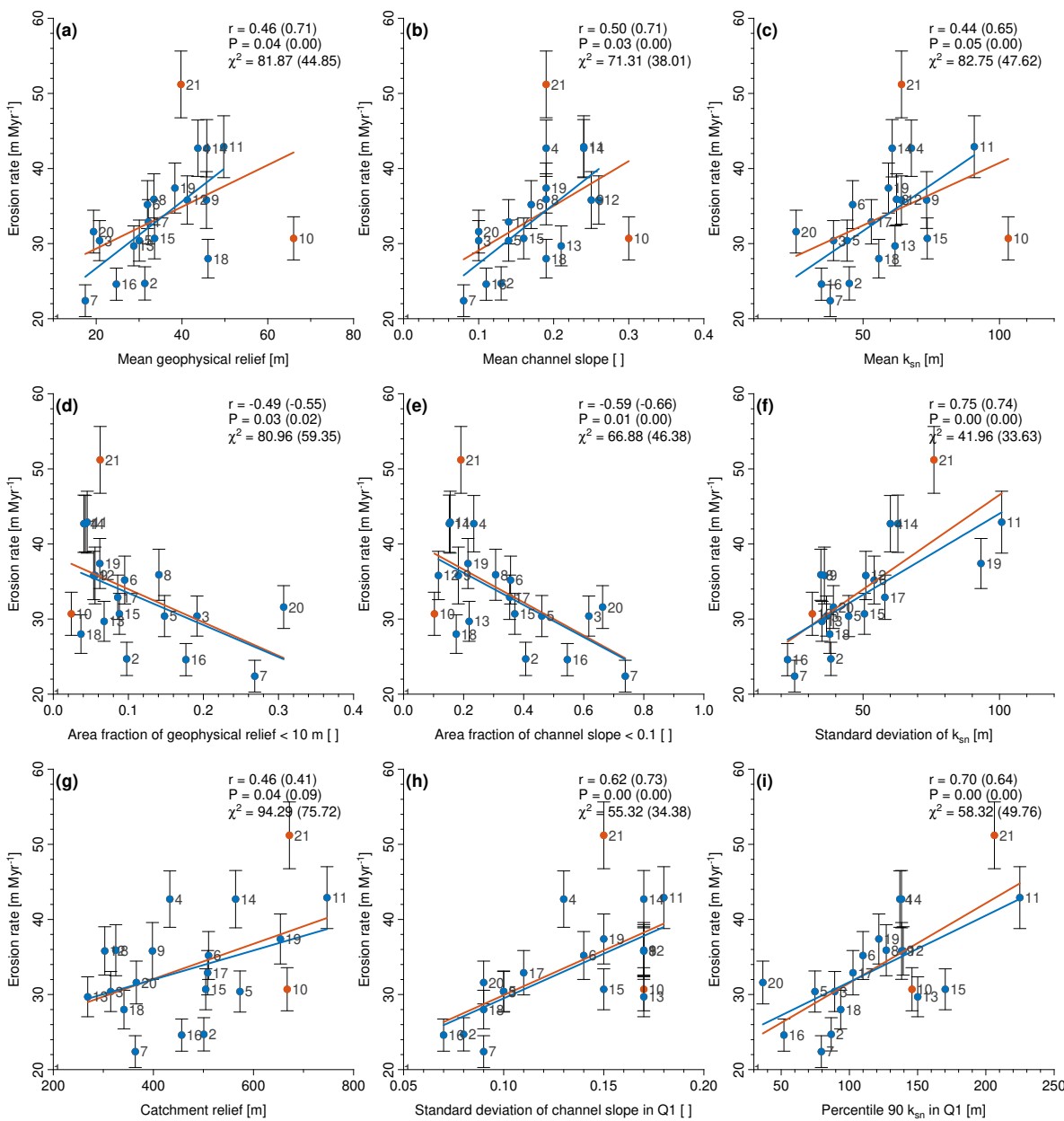

**Figure 7.** Scatter plots showing correlations between erosion rates and topographic metrics. Blue and red dots indicate individual catchments. Red dots mark outlier catchments (C9: Spitzerbach, C20 Kleine Rodl). The best-fit regression line (red line) for all catchments with Pearson's linear correlation coefficient $r$ and its corresponding $P$ value annotated at the top right of each subplot. The blue line represents the regression line excluding the outliers with the corresponding $r$, $P$ and $\chi^2$ values within the brackets.

The normalized steepness index shows larger variations in the degree of correlation compared to geophysical relief and channel slope. Almost all correlation analyses lead to a higher degree of correlation if the outlier catchments are not taken into account. Erosion rate shows a high correlation with the standard deviation and the $90^{th}$ percentile of $k_{sn}$ indicating that erosion rate is controlled over-proportionately by the highest 10% in $k_{sn}$. The distribution of correlation coefficients and various statistical measures for $k_{sn}$ is similar at all elevation quarters, but the degree of correlation decreases significantly towards the highest elevation quarter.

The analysis shows that many statistical attributes of geophysical relief, channel slope and $k_{sn}$ have a moderate to high positive or negative correlation with erosion rate. However, the differences in the degree of correlation are small and may not be significant. This is demonstrated by the (non-)consideration of the outlier catchments, where the degree of correlation of an individual topographic metric and erosion rate changes more than the differences in the degree of correlation between the individual metrics. The comparison of the elevation slices shows that erosion rate has the lowest correlation with the topographic metrics in Q4.

The finding that erosion rate increases with relief and terrain steepness is well known (e.g. Ahnert, 1970; Portenga and Bierman, 2011) but has been extended in this study with novel metrics characterizing entire catchments but also elevation slices accounting for elevation-dependent changes in topography. The correlation analysis between erosion rate and topographic metrics reveals that even in transient landscapes such as the Bohemian Massif with its distinct physiographic transition, catchment-wide topographic analysis allows not only qualitative statements about the erosion rate but also its quantification. In a comparison of neighboring catchments with crystalline basement bedrock, erosion rate increases statistically significantly with geophysical relief, channel slope and $k_{sn}$ and decreases with the proportion of area with low-relief, low channel slopes and low $k_{sn}$.

The error bars in Fig. 7 refer to the uncertainty in erosion rates arising from the internal and external uncertainties according to the relation

$$\delta E = \sqrt{\delta E_{\mathrm{int}}^2 + \delta E_{\mathrm{ext}}^2}. \tag{6}$$

These uncertainties can be used to test the hypothesis that the erosion rate can be predicted by the respective topographic metric based on the straight lines in the diagrams. For this purpose, the property

$$\chi^2 = \sum_i \left( \frac{E_i - E_{\mathrm{pred},i}}{\delta E_i} \right)^2 \tag{7}$$

is considered. Assuming that the errors in erosion rate in different catchments are statistically independent, this property follows a $\chi^2$-distribution with 18 (all catchments) or 16 (without outliers) degrees of freedom, corresponding to the number of catchments minus 2 since a straight line involves two adjustable parameters. All obtained values of $\chi^2$ are even above the 99 % quantile of the respective $\chi^2$-distribution (34.8 for 18 and 32.0 for 16 degrees of freedom), which rejects the hypothesis of predicting the erosion rates in a linear way from one of the topographic metrics at 1 % significance. Practically, this means that the erosion rates cannot be predicted from a single topographic metric. Formal limitations of the analysis (disregarding the uncertainty of the topographic metrics, linear approach, not including the individual uncertainties in the fit, assuming statistical independence) do not affect this result qualitatively.

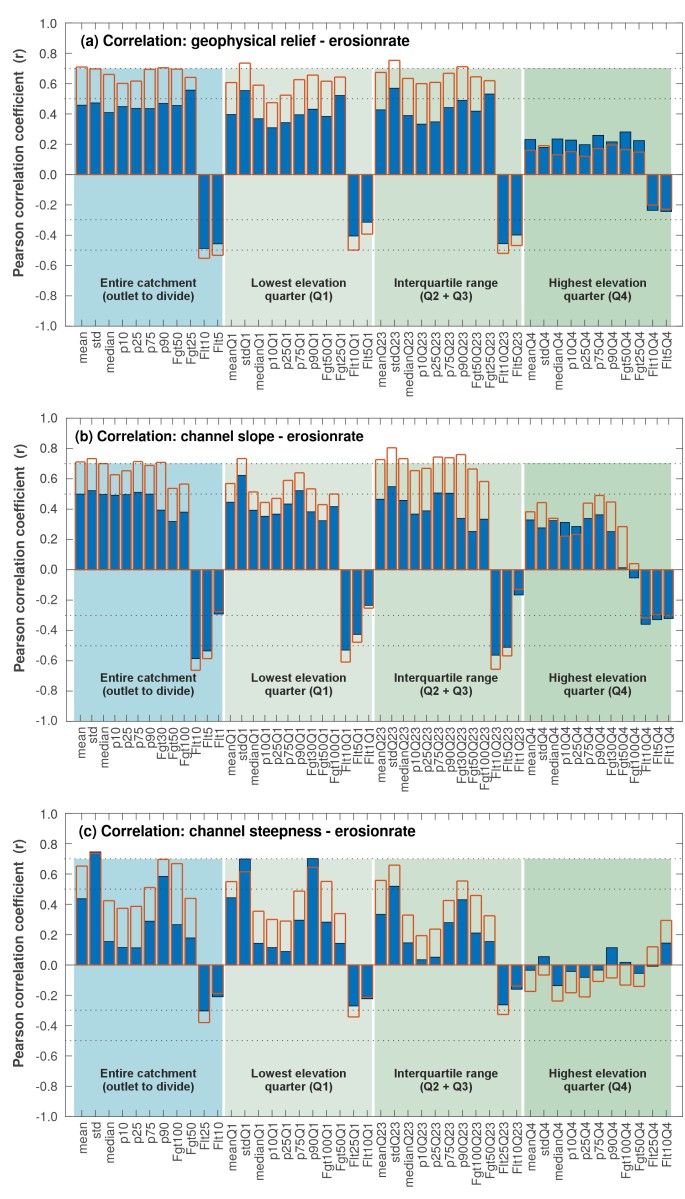

**Figure 8.** Degree of correlation between catchment average erosion rates and topographic metrics of the catchment upstream the sampling location. Blue bars show the Pearson correlation coefficient ($r$) taking into account all 20 catchments and the red bars without the outlier catchments 9: Spitzer Bach and 20: Kleine Rodl.

This finding is in line with the values of $r$ listed in Fig. 7, which are $r \approx 0.75$ at best. Since the $R^2$ value of the regression line is the square of Pearson's regression coefficient $r$, the best obtained correlations yield $R^2$ values not much above 0.5, which means that a bit more than half of the variance of the erosion rates can be explained from the respective topographic metric at best. A multiple regression involving several topographic metrics might improve the prediction in terms of $R^2$ and $\chi^2$ considerably. However, we have to keep in mind that the data set of 20 catchments is quite small. So using a combination of several predictors bears the risk of overfitting and was therefore not taken into account.

Hence, the approach is probably of limited value for estimating erosion rates of individual catchments but might be more suitable at larger scales. Alongside the correlation between topographic metrics and measured erosion rates, the generally low erosion rates despite the occurrence of very steep landforms require an explanation, which is consistent with the topographic evolution of Variscan Massifs such as the Bohemian Massif.

## 4.2 Low erosion rates despite steep landscape patches - A "slowly eroding" topography

Catchment-wide erosion rates between about 20 and 50 $\mathrm{m\,Myr^{-1}}$ for the Bohemian Massif are well in line with the few other data from this region (19 -31 $\mathrm{m\,Myr^{-1}}$) (Dannhaus et al., 2018; Schaller et al., 2001, 2016), and erosion rates reported for catchments of other Variscan Massifs. Considering catchments larger 5 $\mathrm{km^2}$, they are generally low and only exceed 100 $\mathrm{m\,Myr^{-1}}$ on back-eroding flanks of continental rift valleys: Massif Central ($5 - 80$ $\mathrm{m\,Myr^{-1}}$) (Olivetti et al., 2016; Schaller et al., 2001), Black Forest ($26 - 112$ $\mathrm{m\,Myr^{-1}}$) (Meyer et al., 2010b; Morel et al., 2003; Schaller et al., 2001; Wolff et al., 2018) and the Vosges Mountains (34 -88 $\mathrm{m\,Myr^{-1}}$) (Jautzy et al., 2024). However, erosion rates are significantly lower than in the Alps, where high erosion rates ($> 1000$ $\mathrm{m\,Myr^{-1}}$) were predominantly reported for catchments showing glacial imprint (see Delunel et al., 2020, for a review and references therein for details). Erosion rates of the largely unglaciated eastern fringe of the Alps amount to 40 and 150 $\mathrm{m\,Myr^{-1}}$ (Dixon et al., 2016; Legrain et al., 2015) and are thus up to three times higher than those in the Bohemian Massif. Legrain et al. (2015) have shown for the eastern margin of the Alps that erosion rates in catchments with relic planation surfaces are one third of those where such surfaces have been completely dissected.

Evidence from the field work in the gorges of the Bohemian Massif with rivers characterized by high flow velocities and flanks with active mass movements, in combination with the morphometric analyses of the present study and that of Wetzlinger et al. (2023), would suggest erosion rates that are at least similar to those on the eastern margin of the Alps. However, this impression is deceptive insofar as the area fraction of deeply incised gorge with high relief and large $k_{\mathrm{sn}}$ values is small compared to large areas of the catchments that are characterized by long-wavelength topography with low amplitude. Our results are in line with Legrain et al. (2015) and show that the proportion of low-relief surfaces in the entire catchment area has a strong effect on the mean erosion rate. This relationship is clearly demonstrated by the correlations between catchment-wide erosion rates and topographic metrics. Erosion rates increase with increasing $k_{\mathrm{sn}}$ and geophysical relief and decrease with the proportion of elevated low-relief surfaces of the catchments. Thus, the highest erosion rates are found in elongated catchments that essentially drain the steep and incised, Danube-facing escarpment of the Bohemian Massif (i.e., catchments: 10, 13, 20, where canyons dominate the landscape). There, low-relief topography still occurs but is of lower significance compared to other investigated catchments of the region. The two different types of landscape, the steep, deeply incised regions close to the

active receiving stream and the low-relief surface at higher elevations, should have different erosion rates. Catchment areas of sufficient size to largely exclude the influence of stochastic processes (i.e., landsliding) on erosion rate that lie entirely within the steep, incised part of the landscape, do not occur. We measured either a mixed, hard to disentangle erosion signal, to which both low-gradient and incised parts of the landscape contributed, or only the erosion signal of the low-gradient landscapes. We were able to determine the latter by sampling upstream of the prominent knickpoints separating these two contrasting landscape types. Based on the concentration of $^{10}$Be in river sands, we cannot make a clear statement about erosion rates in the lower-lying, steep parts of the landscape. To make statements about this, a nested sampling approach would be necessary where a larger number of samples is taken along a single river. The erosion rates determined in the upper and lower reaches in the Lainitz catchment: P03, P04 and Kleine Mühl catchment: P17, P18 do not allow a conclusive statement to be drawn. However, the distinct correlation between high erosion rates and catchments with deeply incised river landscapes, pronounced relief and steep rivers suggests that the particularly steep low-lying areas near the Danube should have significantly above-average erosion rates.

## 4.3  Soft over hard rocks as a principal control on landscape evolution

Linking the conceptual model (Fig. 3) with key findings from morphometry and cosmogenic nuclide dating (Figs. 6, 7, 8), we suggest that the strong lithological contrast between cover and basement rocks cause landscape transience in the study region. This transience reduces erosion rates, drives topography formation, and in turn influences the evolution of the Central European drainage network (Kuhlemann and Kempf, 2002; Robl et al., 2008, 2017a). Comparable lithological contrasts with soft cover sediments over hard bedrock can be found in all Variscan massifs in Europe (Massif Central, Vosges, Black Forest, Rhenish Massif) (Asch, 2003; Moosdorf et al., 2018; Schaller et al., 2001; Meyer et al., 2010b, a; Morel et al., 2003) or more generally in continental realms where uplift causes a transition from a depositional to an erosional setting (Sobel and Strecker, 2003). In such a setting, ongoing uplift and progressive erosion leads to the denudation of hard basement rocks with the boundary between the contrasting rock types gradually shifting over time and counteracts the establishment of a topographic steady state (Forte et al., 2016; Perne et al., 2017).

The boundary between the Bohemian Massif (hard rocks) and the Molasse Basin (soft rocks) migrates with ongoing erosion southward toward higher sediment thickness. Consequently, the occurrence of catchments that are already completely denuded of the sedimentary cover, those that still show small relicts of it and those that are predominantly covered by sediments show a north to south trend and hence various evolutionary states. Since the response time for adaptation of the channel gradient to the hard bedrock is greater than in the soft cover sediments (e.g. Forte et al., 2016), the adjustment is incomplete and even catchments without sedimentary remnants show the inherited low relief topography of the originally sediment-covered landscape or the shallow contact to the crystalline basement (Figs. 2, 3 and supplemental figures in the Zenodo repository (Robl, 2025)).

Among the investigated catchments, the Aschach catchment is situated at the furthest south of the study area and thus in a region with originally the highest thickness of Molasse sediments (Figs. 1b, 9). At the Aschach catchment, the large-scale occurrence of cover sediments is punctuated by bedrock barriers of crystalline basement that form local base levels (see also

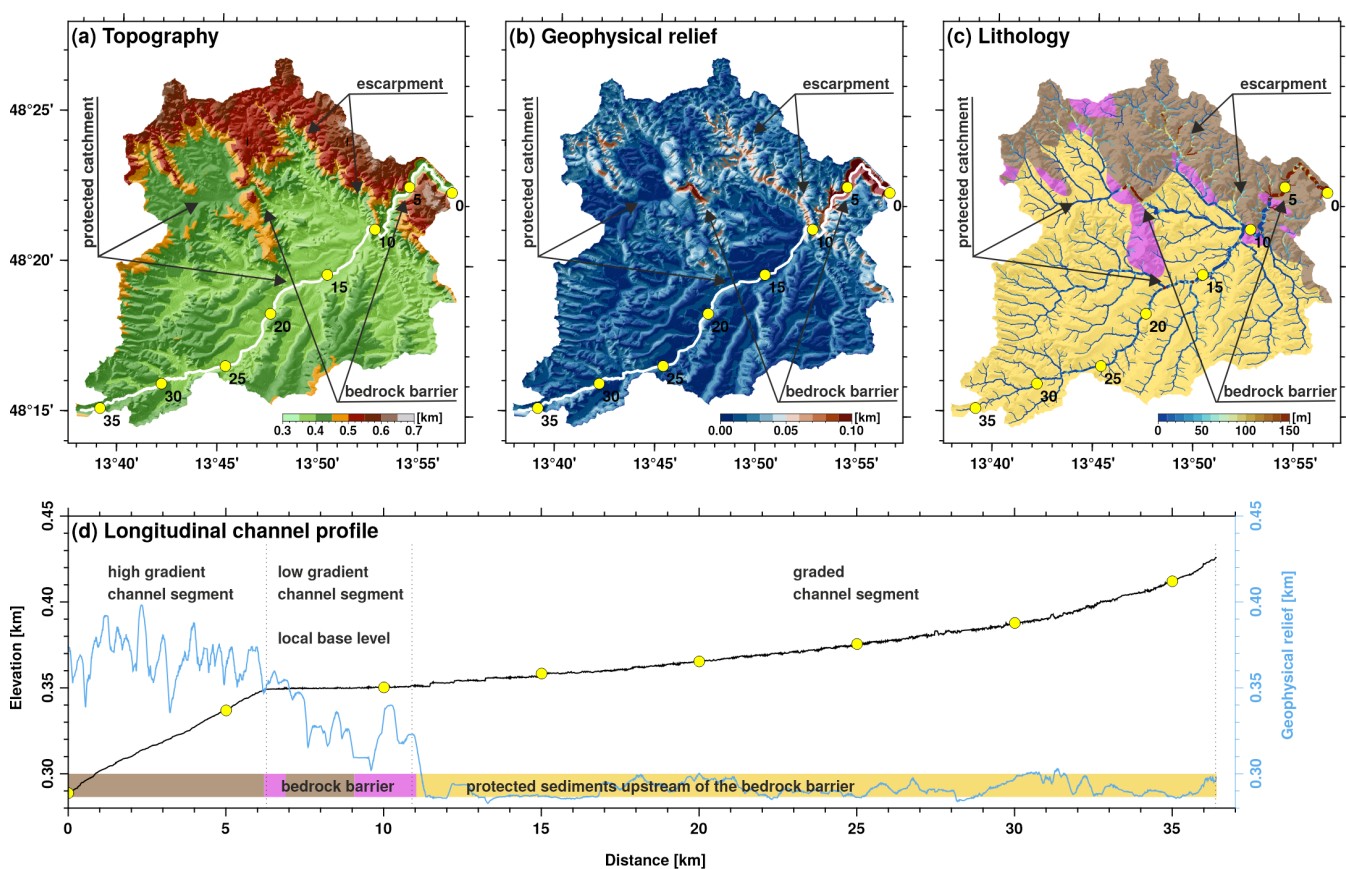

**Figure 9.** Lithology as principal control of relief formation at the transition Bohemian Massif - Molasses Basin (Aschach catchment (P 19)). The main river (white line) and the flow length measured from the outlet to the river source (yellow circles and labels) are shown in the three maps (a) topography, (b) geophysical relief and (c) lithology. In the latter, the river network is color-coded for $k_{sn}$ and the base map shows the predominant lithology. Migmatites (dark brown) and granitic rocks (magenta) in the north belong to the Bohemian Massif and are covered by Molasse sediments (light brown) in the south (see Fig. 1b for a geological map of the study region). (d) Longitudinal channel profile (black solid line) and geophysical relief along the Aschach River (blue solid line). Bedrock types are indicated by the bar at the bottom. The distance markers (yellow circles) can be linked to those in the maps in (a) - (c).

Fig. 3 for explanation). The Aschach catchment is therefore a key region for understanding the topographic evolution of the northern Alpine foreland basin during the last few million years of basin inversion and exemplary for the topographic evolution of an eroding landscape with a layered stratigraphy.

The topography of the Aschach catchment shows an earlier evolutionary state than catchments north of the Danube River and allows investigating how relief formation is controlled by large variations in the bedrock erodibility. Catchment-wide erosion rates slightly exceed $30 \, \mathrm{m \, Myr^{-1}}$ and hence are within the usual range of erosion rates measured in the study region. In turn, mean relief and mean channel steepness are at the lower end of computed values (Fig. 6). However, it is evident that the topographic pattern in the Aschach catchment is dominated by the occurrence of different lithological units. Migmatites and granitic rocks of the Bohemian Massif outcrop at the northern part of the catchment, while sediments of the Molasse Basin occur to the south and occupy the largest part of the catchment (Fig. 9c). Areas with the highest elevation occur in the northern part close to the confluence of the Aschach River with the Danube River and coincide exactly with the occurrence of the basement rocks of the Bohemian Massif, while the headwaters of the Aschach River in the south are characterized by low elevations (Fig. 9a).

The Aschach River enters a gorge roughly $10 \, \mathrm{km}$ upstream from its confluence with the Danube River. There, the range of hills parallel to the Danube valley and the valley flanks of the Aschach gorges consist of basement rocks and are significantly higher than the actual headwater region where sediments of the Molasse Basin outcrop. Such landscape patterns are known from antecedent rivers that have retained their flow direction, although the geometry of the landscape and thus the large-scale topographic gradient have changed substantially. The distinct lithological control indicates that the sediments of the Molasse Basin are more efficiently eroded than the high-grade metamorphic and granitic rocks of the Bohemian Massif. Due to the variable depth of the Molasse Basin with a general increase in sediment thickness to the south, this leads to the sculpting of hill chains from erosion-resistant rocks of the Bohemian Massif in the process of basin inversion, surface uplift and spatial variations in the total amount of erosion. Distinct escarpments (large values in the geophysical relief) have formed at the transition from rocks of the Bohemian Massif to sediments of the Molasse Basin (Fig. 9b). Pronounced relief also arose in the lower reach where the river has deeply incised into the crystalline bedrock by forming a gorge. Such a stepped landscape with low-relief surfaces, separated by escarpments, and canyons, which erode backwards from the escarpments into the low relief surfaces, are common features in the Southern Bohemian Massif.

The longitudinal channel profile is in line with the described plan view characteristics (compare Fig. 9d with the conceptual model in Fig. 3). The steepest channel segment is found in the lower reach where the bedrock consists of migmatites and granites, while the upper course is characterized by low channel steepness (Fig. 9c, d). This is a common characteristic of the rivers that drain the southern Bohemian Massif (Wetzlinger et al., 2023). Interestingly, the prominent knickpoint is not at the transition between the two major lithological units (crystalline basement, sedimentary cover), but within the crystalline basement rocks, suggesting knickpoint mobility and landscape transience. The low channel steepness upstream of the knickpoint is accompanied by low values of geophysical relief suggesting that progressive channel steepening and relief formation in a realm of large-scale surface uplift are linked. Despite flowing in basement rocks, the observed channel steepness upstream the knickpoint corresponds to rivers draining the Molasse zone, where, due to the low resistance of the bedrock to erosion, even

a low channel steepness is sufficient to balance uplift rates by erosion rates. The low channel steepness in the presence of crystalline bedrock indicates that relief formation by the incision of the river into its resistant bedrock occurs with a significant delay after eroding the Molasse sediments – a bedrock barrier emerges and forms a local base level. The low channel steepness as a relic of the river draining the Molasse zone are slowly adjusted to the resistant rocks of the Bohemian Massif. The pace of adjustment is controlled by the migration rate of the knickpoints and hence influences the catchment-wide erosion rate.

We suggest that the occurrence of bedrock barriers is slowing down the erosion of sediments in the upper reaches and reduces the catchment-wide erosion rates, which leads to an increase in elevation (Figs. 9, 3). This can be seen particularly well in a tributary on the orographic left side, which confluences with the Aschach River about 10 km upstream of its outlet and breaks through such a bedrock barrier. The breakthrough is characterized by a large channel steepness and high geophysical relief. The catchment covered by the Neogene sediments upstream of the bedrock barrier is protected against erosion and features a significantly higher mean elevation than neighboring areas within this lithological unit, which makes the catchment prone to be captured by the neighboring river (see Fig.3 for explanation). However, the Aschach River itself with its gorge section in the lower course also forms such a barrier, which influences a much larger area and protects the sediments in the upper course from erosion. At an even larger scale, the breakthroughs of the Danube River, which repeatedly flows from the Molasse zone into the Bohemian Massif and forms deep gorges (Fig. 2), have an even greater influence on the evolution of the upstream drainage network and the topography. The height of the resulting terrain step in the upper reaches of such a lithological barrier depends on the contributing drainage area and the length of the river section within the bedrock. While the former determines the equilibrium channel gradient for the prevailing bedrock and uplift rate, the latter yields the total height of the terrain through the simple geometric relationship of terrain step height with increased flow gradient and distance.

As a consequence of the reduced erosion rate upstream of emerging bedrock barriers, a stepped landscape forms. This process can explain the formation of stepped landscapes due to lithology contrasts without repeated changes in the uplift rate or base level for which evidence is usually lacking. The emergence of local base levels at different elevations due to bedrock barriers makes low-gradient rivers at higher elevation prone to being captured. The observation of T-shaped river junctions, 90 degree bends in river courses and wide abandoned valleys with wind gaps indicate a lithology-controlled reorganization of the river network in the region (Figs. 1a, 3). In particular, the wide, abandoned valley west of the city of Krems, which runs parallel to the actual Danube valley and contains fluvial sediments assigned to the Alpine region (Österreich, 2021; Nagl and Verginis, 1987), indicates that the course of the Danube River was shifted south as a result of the process described above.

## 5    From an alluvial plain to a low mountain range

Using the example of the Aschach catchment (Fig. 9) and in line with the conceptual model of landscape evolution of the Bohemian Massif – Molasse Basin region (Fig. 3), we can show that the transition from sedimentary cover to crystalline basement and the occurrence of bedrock barriers in river courses controls the formation of topography upstream. The Aschach catchment with its extensive sedimentary cover shows an earlier stage of landscape evolution compared to the catchments further north, where the original thickness of sediments was much smaller and the bedrock of the rivers consists now predominantly of crys-

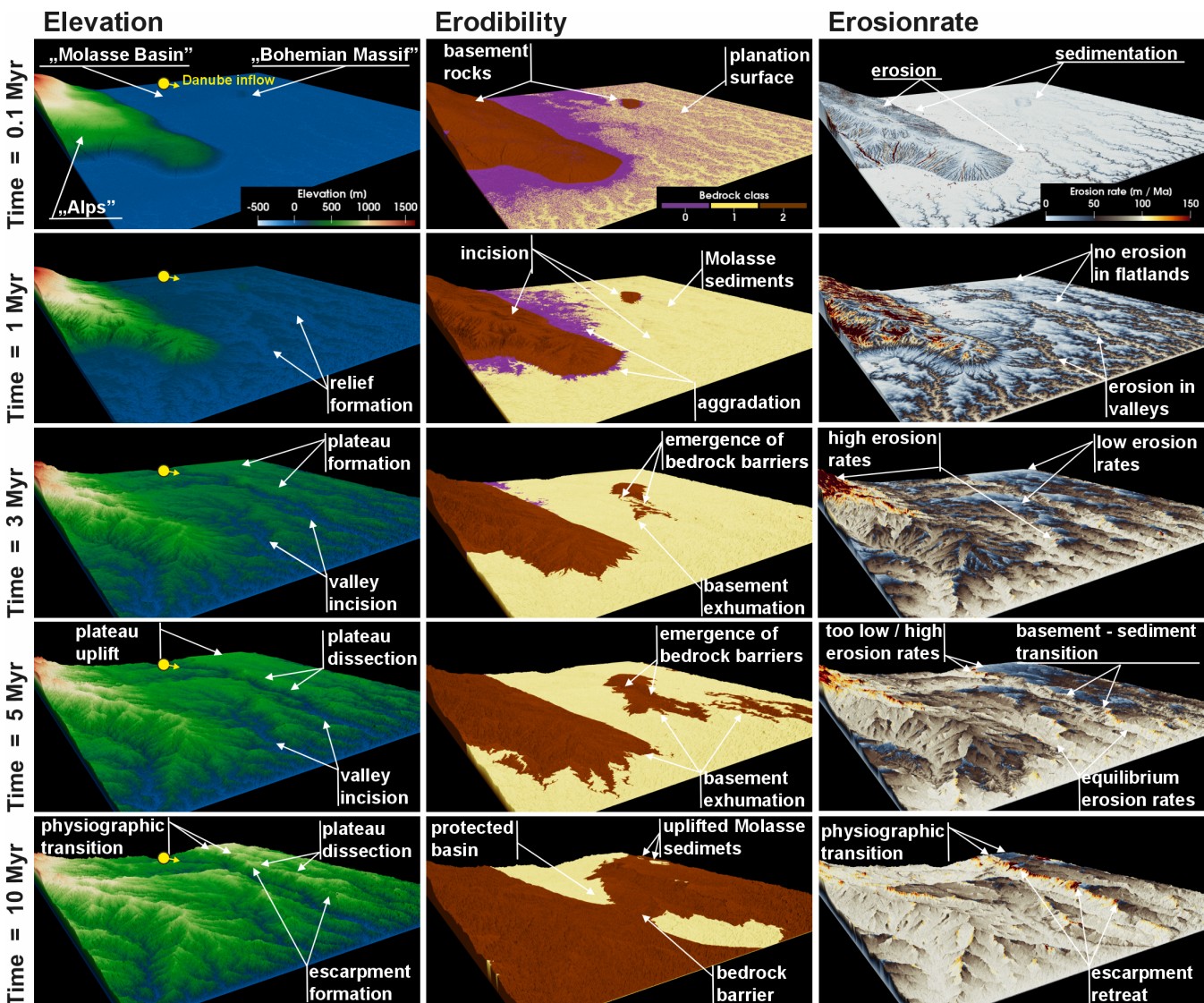

**Figure 10.** Time series of a landscape evolution model exploring the influence of contrasting substrate properties on topography formation under uniform uplift. See also supplemental videos at the Zenodo repository (Robl, 2025).

talline basement. Based on a numerical model, we describe stages of relief formation from an alluvial plain to a low mountain range driven by uplift and distinct bedrock contrasts. Thereby, we test our hypothesis that the first-order topographic features of the study region result from the erosion contrasts between the sediment cover and the crystalline basement at uniform uplift rates (Fig. 10). The model is similar to the approach of Forte et al. (2016), but it takes into account the region-specific geological situation (i.e., the basin geometry and principal flow direction) and sediment transport.

The modeled evolution of the landscape (Fig. 10 and supplemental videos (Robl, 2025)) commences with extensive planation
surfaces that drop gently from west to east with a difference in elevation of 20 m. Significant topography with peak elevation
exceeding 1500 m exists only in the realm representing the Eastern Alps.

The first 0.1 million years after the onset of the basin inversion are characterized by the initial evolution of the drainage
system. At that time, only the Alps and a small part in the realm of the Bohemian Massif are characterized by basement rocks.
Large parts of the still flat lying Neogene sediments of the Molasse Basin are already covered by fluvial sediments. These
sediments originate primarily from the erosion of the Alps and subordinately from a topographic high in the region of the
Bohemian Massif, which towers above the alluvial plain. In the plain areas of the Molasse Basin, the sediment flux exceeds the
transport capacity of the rivers, resulting in large-scale sedimentation. At the same time, river networks are beginning to incise
progressively from the slightly lower eastern model boundary. The pre-defined river representing the Danube River enters the
model domain at the western border and dominates the fluvial erosion of the entire area.

The formation of valley-ridge line relief both in the Alps and in the alluvial plain of the Molasse Basin becomes visible
already after 1 Myr. By adjusting the channel gradients to the prevailing uplift rate and substrate properties, the sediment
is effectively transported through the drainage network. Only the transition between the Alps and the Molasse Basin is still
flanked by fluvial deposits. While the initial relief in the Alps causes erosion rates to be significantly higher than the uplift
rate of $100 \, \mathrm{m \, Myr^{-1}}$, a rough balance between uplift and erosion has already been established in the larger rivers draining
the Molasse Basin. Between the incised valleys, however, there are large areas with still low topographic gradients where the
erosion rate is close to zero. This in turn leads to an increase in elevation and the formation of elevated low-relief surfaces.

Even after 3 Myr of uplift, the landscape is still dominated by incised valleys and intervening plateaus. While the area of
the plateaus declines due to headward erosion, the relief between the plateaus and valleys increases, which gives the landscape
a bimodal appearance with a distinct physiographic transition separating elevated low-relief surfaces from incised landscape
patches. These contrasts in the landscape are intensified by progressive erosion and the exposure of basement rocks of the Bo-
hemian Massif where previously easily erodible rocks of the Molasse Basin occurred. The lithological changes, both spatially
and temporally, lead to the first emergence of bedrock barriers and formation of escarpments between the river valleys incising
into basement rocks and the adjacent plateau areas.

After 5 Myr, the large rivers including the "Danube River" have at least partly reached the basement rocks. This results
in the formation of major bedrock barriers, which act as local base levels and thus distinctly slow down erosion in the upper
reaches. The imbalance between erosion rates in the valleys and plateaus leads to further relief increase despite uniform uplift
rates. Due to the increasing exposure of basement rocks and the large variations in catchment size, there are large spatial
differences in the degree to which a topographic steady state has been approached. Furthermore, the state of the topography
changes abruptly once a new bedrock barrier is formed. In the northwestern part of the model area, for example, there is still
an extensive plateau that breaks off steeply towards the Danube valley. But even on the eastern margin of the Alps, where the
bedrock is increasingly being exposed by the erosion of Neogene sediments, the erosion rates cannot compensate for the uplift
rates and the topography is growing.

After 10 Myr, a large fraction of the Neogene sediments has already been eroded and is only exposed in two areas separated by major bedrock barriers. In large parts of the model domain, fluvial erosion compensates for uplift. However, an escarpment several tens of kilometers long and up to 1 km high has formed on the orographic left side of the Danube valley, separating low-relief surfaces still covered by Neogene sediments in the north from the incised Danube valley in the south. The rivers that originate on the plateau either drain eastwards at a low gradient or directly into the Danube. The latter group is characterized by a strong physiographic transition, where the low gradient of the rivers on the plateau turns into a high gradient at the transition to the escarpment. The distribution of erosion rates shows that the escarpment features largely equilibrium erosion rates. The adjacent plateau is still characterized by low erosion rates. At the plateau edge, however, erosion rates occur repeatedly well above the equilibrium rate and indicate the reorganization of the drainage network through headward cutting rivers. Peak erosion occurs whenever a low gradient river of the plateau is captured by the steep rivers of the escarpment - a process that should also be significant for the study region and for which there are several indicators in the topography such as T-shaped river junctions, or elbow shaped bends in rivers (Wetzlinger et al., 2023).

The model shows that bimodal landscapes with distinct physiographic transitions between elevated areas of low-relief and low-lying areas that are steep and dissected by rivers arise in a geological setting with a uniform uplift rate due to strong variations in the erodibility of the rocks. The observed topographic features in the Bohemian Massif, which we have elaborated in detail using the Aschach catchment and are indicative of the entire region, are given a temporal dimension by the model. The model also shows that, due to the transient state of the topography, the catchment average erosion rates only represent snapshots on the trajectory towards a topographic steady state. The individual catchments have approached this state to varying degrees and the topographic state can change again abruptly after a river piracy event. In the aggressor catchment, erosion rates are distinctly increased downstream the capture point but may be temporarily reduced on catchment average if the catchment area is extended by low-relief surfaces.

The long-term uplift rates of the region and erodibility of the different rocks are poorly constrained. So the presented model results only show one possible scenario for timing and rates of topography build-up in the Southern Bohemian Massif. Higher uplift rates would lead to a stronger expression of landscape bimodality. Smaller bedrock erodibilities would have a similar effect as larger uplift rates. A further increase of the erodibility contrast from 1:10 between crystalline basement of the Bohemian Massif and the Neogene sediments of the Molasse Basin leads to similar topographic patterns with slightly more pronounced escarpments (see also supplemental movies for larger contrasts in rock erodibility (Robl, 2025). With significantly smaller erodibility contrasts, the landscape bimodality is mainly due to fluvial prematurity and vanishes towards morphological equilibrium. However, the morphology of the Aschach catchment shows that both hillslopes and rivers are significantly steeper in the crystalline basement than in the Molasse sediments, supporting the assumption of pronounced differences of the bedrock in resistance to erosive surface processes.

# 6 Conclusions

In this study, we correlated catchment-wide erosion rates of 20 rivers of the Southern Bohemian Massif with topographic metrics representing both the hillslope and the drainage system domain of the investigated catchments. By confronting these findings with results from a numerical model describing the topographic evolution in a slowly uplifting region with strong lithological contrasts we come to the following conclusions.

- Erosion rates between 20 and 50 $\mathrm{m\,Myr}^{-1}$ in the Southern Bohemian Massif are in line with reported values from other
Variscan Massifs but substantially lower than in the unglaciated eastern fringe of the Alps.

- The occurrence of deeply incised and steep river segments with active hillslopes near the receiving streams (Danube River, Vltava River) seem to contradict the low erosion rates. However, substantial areas of the investigated catchments are occupied by elevated low-relief surfaces that lower the erosion rate on a catchment average.

- The correlation of erosion rates with topographic metrics demonstrates distinct positive correlations between erosion
rate with channel steepness and with geophysical relief showing the influence of both fluvial and hillslope processes on catchment-wide erosion rate. The lowest erosion rates occur in catchments with a large area fraction of low relief.

- The degree of correlation between erosion rate and landscape geometry is greater in the lowest elevation quarter (Q1) than in the highest elevation quarter (Q4), which indicates an increasing decoupling of topographic properties and erosion rate with distance from the sampling location for cosmogenic nuclides in river sands.

- The morphological analysis and a time-dependent numerical model support the hypothesis that the evolution of the landscape of the study region is controlled by erodibility contrasts between erosion-resistant rocks of the Bohemian Massifs overlain by the easily erodible rocks of the Molasse Basin. The emergence of uplifted low-relief surfaces separated from a deeply incised landscape by a pronounced physiographic transition is enhanced by strong lithology contrasts.

- The emergence of bedrock barriers in rivers with progressive river incision results in distinct escarpments and planation
surfaces at different elevation levels despite uniform uplift. Spatial or temporal variations in uplift rate due to faulting are not required to explain a stepped landscape and a strong landscape diversity during relief rejuvenation.

- The high erosion resistance of bedrock barriers prevents the erosion of easily erodible rocks (i.e., "Neogene Molasse Basin" sediments) in the upper reaches and thus controls the catchment-wide erosion rate.

*Code and data availability.* All data for computing catchment-wide erosion rates are included in the article. The time-dependent model
for describing the landscape evolution in the domain of the Bohemian Massif – Molasse Basin can be found at the Zenodo repository: https://doi.org/10.5281/zenodo.14998534 .

*Video supplement.* The videos supplement Fig. 10 and show the time-dependent evolution of the topography, bedrock properties and erosion rates in the study region. The videos show the landscape evolution for an erodibility difference ($K_d$) between basement rock of the Bohemian Massifs and the sediments of the Molasse Basin of 1:10 (as shown in Fig. 10) and 1:100 and can be found at the Zenodo repository: 665 https://doi.org/10.5281/zenodo.14998534 .

*Author contributions.* J. R., F. D., K. S. and C. vH. prepared the study design, J. R. and K. S. did the field work, D. F. and K. S. were responsible for sample preparation and analysis of cosmogenic nuclides. J. R. and F. D. carried out the morphometric analyses and their statistic evaluation. J. R. and S. H. developed the landscape evolution model for the study region. J. R. created the figures and videos. The contribution to the writing of the text corresponds to the order in the list of authors.

*Competing interests.* The corresponding author confirms that none of the authors has any competing interests.

*Acknowledgements.* This research has been financially supported by the Austrian Academy of Sciences and the Austrian Federal Ministry of Education, Science and Research as part of the initiative "Earth System Sciences Research Programme". The open access publication was supported by the Paris Lodron University of Salzburg Publication Fund.

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
