# Peer review of "Old orogen – young topography: lithological contrasts controlling erosion and relief formation in the Bohemian Massif"

_EGUsphere, 2024_

## Author Comment (AC1)

**Authors' Response to Reviews of**

**"Old orogen - young topography: Lithological contrasts controlling erosion and relief formation in the Bohemian Massif"**

Jörg Robl et al.
*ESURF,*

RC: *Reviewers' Comment*,     AR: Authors' Response

Dear reviewer,

Thank you for taking the time to evaluate our manuscript and for providing valuable suggestions for improvement. We are pleased that the review did not question the applied methods and results. However, after reading the review, we realized that we had overlooked several structural problems in the presentation. We think that this is also the reason why several aspects were understood differently from what we wanted to present. We are confident that we can solve the problems mentioned and are ready to improve the manuscript accordingly so that it is suitable for publication in ESURF.

**Raised issue:**  The first major point of criticism is the 'inconsistency between the goal of this study and the presentation of the results'. After carefully reading the article again from this point of view, we can understand the criticism. This problem, as well as other major issues described in the review, arise from the introduction, where the research question is not stated clearly enough.

**Planed task 1:**  We will rewrite the introduction and explain the research question in general terms before we characterize our study region: Our aim is to understand the effect of strong erodibility contrasts between the cover sediments and the crystalline basement on the evolution of topographic patterns and erosion rates during inversion (uplift) of a peripheral foreland basin. A situation with easy-to-erode sediments overlying hard-to-erode bedrock, where rivers remove the cover sediments and reach the bedrock with progressive uplift, can be observed on the periphery of many mountain ranges and leads to characteristic topographic patterns.

**Planed task 2:**  We will take up the suggestion of reviewer 2 and draw a cartoon to accompany the new introduction on the evolution of the landscape in such a geological setting.

**Raised issue:**  As a part of the first major point of criticism, the lack of separate morphometric analyses for different bedrock types and for different landscape geometries is criticized. Our focus on simple catchment-averaged topographic metrics is motivated by the fact that erosion rates based on the concentration of cosmogenic nuclides are also averaged and represent a mixed signal of different types of rocks and landscape geometries. In general, this signal can hardly be disentangled either. A splitting of the regions according to topographic features (e.g. manual mapping of low relief surfaces) or the splitting of the regions into different bedrock types makes little sense in this respect.

**Planed task 1:** Explanatory paragraphs at the end of the introduction and in the description of the correlations between erosion rate and catchment-wide metrics explaining why catchment-wide topographic metrics are useful in this study.

**Planed task 2:** A better integration of the figures and the extensive table in the supplement to demonstrate differences in topographic metrics between catchments dominated by the crystalline basement and those dominated by the sedimentary cover (Aschach catchment).

**Planed task 3:** A better integration of and referencing to the pilot study we conducted (Wetzlinger et al., 2023), where a detailed morphometric analysis is shown. However, showing the figures of the spatial distribution of the topographic metrics again in this paper seems not be be very useful to us.

**Planed task 4:** Accompanied by a new figure (cartoon), we will explain in the introduction, which topographic patterns we expect at the transition from sediment to crystalline bedrock and how these patterns change over time: the transition from sedimentary to crystalline bedrock leads to transient conditions in both the fluvial channels and the hillslopes. Consequently, the topographic metrics within crystalline bedrock-dominated regions show a high degree of variance, as we also show in our figures. The change from cover sediments to basement rocks will continue to have an effect for a long time as erosion progresses until the river geometry has adjusted to the new substrate properties. This can be seen very clearly in the Aschach catchment, where the knickpoint in the longitudinal river profile is located within the basement bedrocks and not further upstream at the transition from basement to cover rocks. The removal of the sedimentary covers down to the basement is significantly faster than establishing an equilibrium longitudinal river profile in the basement bedrock. The long response time for the landscape to adapt to the change in lithology is responsible for the marked variations in topographic metrics within the crystalline basement.

**Planed task 5:** Explain, why we think that the area fraction of topographic metrics below / above thresholds in relief and channel slope are better suited for correlation analysis than expert-based mapping: the low relief surfaces mentioned are not a plateau but areas with a long wavelength–low amplitude topography. Expert-based mapping is possible in principle, but it is much less reproducible than determining the area fraction of catchments with topographic metrics (e.g. relief, steepness index) that exceed or fall below defined thresholds. We determined the topographic metrics for the different elevation levels as the signal of landscape adjustment to new conditions migrates upstream (i.e. steepening the channel and increasing local relief). These metrics are also included in the table in the supplement. We also think that the area fraction of topographic metrics below / above thresholds in relief and channel slope shows very well the respective proportions of the two landscape types in each catchment.

**Planed task 6:** Include a geophysical relief map to the supplement to show the spatial distribution of the two landscape types (incised / low relief) on a map. We agree that it would be easier for the reader to follow the story if the physiographic transition separating the incised from the low relief surfaces is shown. We will plot the physiographic transition as a line on the topographic map in current figure 1 and in the supplemental geophysical relief map.

**Raised issue:** It was difficult to understand why the Aschau catchment is of particular importance for understanding the landscape evolution of the region.

**Planed task:** The new introduction including the cartoon figure will solve this problem. The Achach catchment is the only catchment in the study region with a significant amount of cover sediments. However, the other catchments were also (at least partly) covered by these sediments at an earlier evolutionary stage. Due to the greater sediment thickness in the Aschach catchment (closer to the Alps than the other catchments), the evolution of a catchment controlled by sedimentary bedrock to a catchment controlled by crystalline basement rocks is delayed. The Aschach is therefore a key catchment for understanding the landscape evolution of peripheral foreland basins during basin inversion. This is the reason why we took a closer look at this catchment in the discussion section. However, we agree that the term 'representative catchment' is not appropriate.

**Raised issue:** The second problem mentioned in the review concerns the organization of the manuscript. In particular, the section on the Aschach catchment and the description of the numerical model were identified as misplaced. We understand that the organization of the manuscript is not well received by all readers. In

principle, ESURF allows for both the classic structure with introduction, method, results and discussion or a more narrative structure where the method, results and discussion are treated together.

**Planed task 1:** In the revision, we would like to follow the less restrictive version and further relax the classic manuscript structure towards a more narrative style. We will break up the results and discussion sections and write new headings and transitions between sections.

**Planed task 2:** We plan to present the Aschach catchment much earlier in the manuscript (prior to the correlation analysis) as a key catchment for landscape evolution. This also gives us the opportunity to explain how mixed erosion and topographic signals are composed when viewed in the catchment average.

**Planed task 3:** The numerical model was designed to describe the key features of relief formation in an uplifting foreland basin. It is not about reproducing the exact timing and rates of landscape change, as about showing the evolutionary stages of a landscape in a region characterised by uniform uplift and strong spatial and temporal differences in bedrock properties. The exact determination of the bedrock erodibility is of secondary importance as long as the values lie within a reasonable range. We will give references for that and we will discuss the impact of different erodibility values (K) and erodibility ratios even more clearly than before.

**Raised issue:** Apparently confusing statements about spatially and temporally constant uplift rates were mentioned a the third point. We think that this is just a minor misunderstanding.

**Planed task:** It is clear that the inversion of the foreland basin leads to a change in the uplift rate, as a result of which the topography undergoes a transient state. If we speak of the formation of terrain steps despite uniform uplift, we mean uniform uplift with onset of basin inversion. The stepped landscape emerges as a result of the varying properties of the bedrock in a rising landscape. We will clarify this by revising the introduction.

**Raised issue:** In the last major issue, a lack of proper contextualization was addressed.

**Planed task:** We will explain the broader context in the introduction and discuss it at the end of the manuscript. There are numerous studies dealing with the influence of bedrock properties on erosion rates and landscape characteristics. However, as far as we know, there are no studies that address the topographic evolution of peripheral foreland basins after the mountain building phase. Peripheral foreland basins accompany all major mountain ranges on Earth and the findings of this study can be applied to other comparable regions.

**Line-by-line comments**

We will be happy to address the line-by-line comments in the course of a revision – thanks again for that!

On behalf of the co-authors

Jörg Robl

---

## Author Response (AR1)

**Authors' Response to Reviews of**

**"Old orogen - young topography: Lithological contrasts controlling erosion and relief formation in the Bohemian Massif"**

Jörg Robl et al.
*ESURF,*

RC: *Reviewers' Comment*,     AR: Authors' Response

Dear editor, Dear reviewers,

Thank you for taking the time to handle and evaluate our manuscript and for providing valuable suggestions for improvement. We are pleased that the review did not question the applied methods and results. However, after reading the review, we realized that we had overlooked several structural problems in the presentation. We have addressed the issues raised and revised the manuscript according to the reviewers' comments. Among other things, we have substantially rewritten the introduction, added a new section (section 2) explaining our conceptual model, and completely revised the results/discussion section by presenting additional explanations on the significance of bedrock changes, expanding the statistics on the correlation between erosion rates and topographic metrics, and strengthening the contextualization. We have not only added new figures (Figs. 2, 3) but have also improved existing figures (Figs. 1, 6, 7, 9) in line with the reviewers' comments. We are confident that we solved the issues mentioned and that the improvement make the article suitable for publication in ESURF.

**1. Review 1**

Reviewer 1 has identified several issues in the first submission of our article: (a) consistency between the goal of this study and the presentation of the result, (b) organization of the manuscript, (c) confusing statements about uplift and (d) contextualization - importance of the study. We have spent a great amount of time on solving these issues and are confident that we have found very good solutions with an improved structure of the article, a more detailed introduction, a new section on the research hypothesis, new illustrations and clearer wording.

**1.1. General Comments**

RC: *The present manuscript discusses causes of the bimodal landscape in the Bohemian massif using geomorphic indices, basin-averaged erosion rates, and landscape evolution model. The topography in the study area consists of elevated low-relief surface upstream and deeply incised valley downstream, which roughly correspond to areas of metamorphic & granitic rock units and sedimentary rock units, respectively. The authors argue that this contrasting topography resulted from the difference in rock erodibility rather than transient response of the massif to the changes in uplift rates. I think this could be a good case study that demonstrates the controls of bedrock erodibility on landscape evolution and helps to understand the evolution of the bohemian Massif. In particular, I value that the authors combined the DEM analysis, 10Be-based erosion rates, and landscape evolution model. Although the manuscript would be of interest to the readers of ESurf, there are critical issues in the current manuscript that require significant revision of analytical methods, manuscript organization, the validity of the conclusion, and the contextualization. I provide major comments followed by line-by-line comments below, which I hope to help improving the*

*manuscript.*

AR: Thank you for this comment. It shows us very well that we didn't do a particularly good job in the original introduction. We have rewritten the introduction and now guide the reader to our hypothesis on the landscape evolution of the region by providing well-structured information on the region (geological history, morphological features, lithology contrasts, constraints on uplift rates) and explaining how soft over hard rocks affect landscape evolution.

 In particular, we do not argue "that this contrasting topography resulted from the difference in rock erodibility rather than transient response of the massif to the changes in uplift rates". Our study regions is located in the realm of an asymmetric foreland basin, which was formed by the flexure of the lithosphere as a result of the thickening of the crust in the area of the Alps. This region was characterized by subsidence and (marine) deposition over several million years, and it was only the inversion of the basin (uplift) that led to the formation of mountain topography. However, our study shows that the geometry of the topography is strongly dependent on the variable thickness of the sedimentary cover. When we speak of uniform uplift, both spatially and temporally, we mean an uplift event that started with the onset of basin inversion and does not exhibit large spatial and temporal gradients. Various levels of low relief surface have often been interpreted in the past in terms of different uplift pulses (strong temporal variations in uplift rate), something that is difficult to explain by tectonic processes. We show that contrasting bedrock properties (i.e. the bedrock barriers) allow the emergence of these different levels of low relief surfaces without dramatically changing the uplift rate in either space or time. This mechanism provides an alternative explanation for stepped landscapes in the Variscan massifs of Europe but also in the Eastern Alps.

 To clarify, we have rewritten and restructured the entire introduction to clarify the scope of the study. Reviewer 2 suggested that we could draw a cartoon to explain our model for topographic evolution, which we have done. In turn we have introduced a new section (2), which explains the effect of soft over hard rocks on topography evolution and erosion rate.

RC: *The first issue is the inconsistency between the goal of this study and the presentation of the results. Although the goal of this study is to test the hypothesis that the current landscape is largely controlled by rock type, the current manuscript does not include statements nor figures that show the difference in geomorphic indices or erosion rates between rock type. Also, the manuscript does not provide information regarding the position of the elevated plateau and incised valleys, thus it is very hard to evaluate the contrasts of geomorphic indices and erosion rates between them. Filtering the topographic data according to percentile values of elevation in each catchment may help to see the difference between the elevated plateau and incised valleys, but I wonder why the authors did not analyze them separately when calculating geomorphic indices, which must be more straightforward. Also, it is difficult to understand why the Aschach catchment is representative of the study area. In most of the studied catchment, metamorphic & granitic rocks occur in the upstream part while sedimentary rock occurs downstream. This is clearly not the case in the Aschach catchment.*

AR: **Raised issue:** The first major point of criticism is the 'inconsistency between the goal of this study and the presentation of the results'. After carefully reading the article again from this point of view, we can understand the criticism. This problem, as well as other major issues described in the review, arise from the introduction, where the research question is not stated clearly enough.

AR: **Performed task: revise introduction** We have rewritten the introduction section to better prepare the reader for the working hypothesis and the objectives of the study. The section is considerably longer and is divided into new subsections. We have removed the section study site. The subsections deal with the geological evolution of the region with a focus on the superposition of Neogene sediments on the crystalline basement

and on the Neogene uplift, the morphological characteristics as already described in detail in (Wetzlinger et al., 2023) and the effect of rocks with strongly varying resistance to erosion on topography and erosion rates. Based on this, we have designed a conceptual model of landscape evolution (including a new figure), which has been given its own section and leads to the working hypothesis that we are testing in the study.

AR: **Performed task: sketch explaining landscape evolution** We have drawn a sketch that shows a conceptual model of the topographic evolution of an eroding landscape with strong lithological contrasts and thus illustrates our working hypothesis. Please see new figure 3. This should make it clear that computing topographic metrics in a geological setting where soft sediments overlay hard basement rocks is of little informative value.

AR: **Raised issue: morphometry** As a part of the first major point of criticism, the lack of separate morphometric analyses for different bedrock types and for different landscape geometries is criticized.

AR: **Performed task: Explain reasons for the chosen approach** We explain why morphological analyses separately for cover and basement rocks are not useful in the study region. Therefore we have added an explanatory paragraph at the end of the introduction section (subsection: Effect of lithology on eroding landscapes) and section 2 describing the conceptual model of landscape evolution.

AR: **Performed task: new explanatory paragraph** We have added an explanatory paragraph at the beginning of the results/discussion section explaining why we use catchment-wide topographic metrics.

AR: **Performed task: new morphometric figures in the supplement** The supplement contains a new figure in the style of fig 1 showing the geophysical relief characterizing the hillslopes and the drainage network color-coded for $k_{sn}$. From our point of view, the two different landscape types (low relief, incised) are clearly identifiable here. We have refrained from drawing lines for the physiographic transition because the many lines along the incised rivers make the illustrations very difficult to read. Beyond that we provide detailed morphometric analyses for all 20 catchments. We refer to these figures in the first paragraph of the result and discussion section. This is for readers who are interested in the morphological details. However, these figures are not necessary for understanding the study.

AR: **Performed task: new figure showing landscape bimodality** In order to get an impression of the landscape without having to consult the supplement, we have added a new figure (Fig. 2) showing a photo of a Danube river channel in the bedrock and the position of low relief surfaces as well as a perspective representation of the topography of this region with corresponding labeling.

AR: **Performed task: better referencing** We refer at several suitable positions to the pilot study (Wetzlinger et al., 2023), which provides a detailed description of the morphology of the study region. However, we do not think it would be beneficial to repeat the analysis results from this work.

AR: **Raised issue: importance of the Aschach catchment** It was difficult to understand why the Aschau catchment is of particular importance for understanding the landscape evolution of the region.

AR: **Performed task: revision of introduction, new section 2, better contextualization** The new introduction and section 2 including the cartoon figure explaining the landscape evolution should solve this issue. In addition, we have revised the section on the Aschach catchment (subsection: Soft over hard rocks as a principal control on landscape evolution), placing the morphological features in a broader context. The Aschach serves as an example to explain how an eroding landscapes evolve in a soft over hard rocks setting.

RC: *The second issue is about the organization of the manuscript. The current discussion section contains paragraphs that are clearly about "method" or "results". For instance, the section 5.2 explains the*

*relationship between topography and bedrock lithology in the Aschach catchment. However, there is little explanation about this catchment in results, and all the information appear in this section. The section 5.3 includes model setup and the results of the model outputs. The model setup has to be in the method section, and the model outputs should be in the result section.*

AR:  **Raised issue: organization of the manuscript** The second problem mentioned in the review concerns the organization of the manuscript. In particular, the section on the Aschach catchment and the description of the numerical model were identified as misplaced. We understand that the organization of the manuscript is not well received by all readers. In principle, ESURF allows for both the classic structure with introduction, method, results and discussion or a more narrative structure where the method, results and discussion are treated together. However, we agree that we have failed to guide the reader through the study at several points and that there were illogical breaks in the structure of the manuscript.

AR:  **Performed task: restructuring the manuscript** We reorganized the manuscript by relaxing the classic manuscript structure towards a more narrative style. We combined the results and discussion sections and implemented new headings and transitions between sections.

AR:  **Performed task: wider context** We have decided to leave the part describing the Aschach in its original place. However, we are now placing this section in a larger context of lithological control of landscape evolution (layered stratigraphy) through the revised first paragraphs. We also make cross-references to the conceptual model of landscape evolution (Sect. 2). We explain that the transition from the Bohemian Massif to the Molasse Zone and the Aschach catchment in particular is exemplary for many regions and describe the topographic features of the catchment and their significance in this wider context.

AR:  **Performed task: shift sections** As suggested, we have shifted the Model Setup to the method section.

RC:  *The third issue is about the conclusion. Although the authors argue "The resulting stepped landscape requires neither spatial nor temporal changes in uplift rate but can form by erodibility contrasts under uniform uplift conditions", the current topography of the Bohemian Massif seems to be undergoing a transient response to uplift that formed the Massif. As the authors replicated in their landscape evolution model, the current topography of the Massif cannot be formed without the initiation of this uplift. It was very confusing to me that the authors argue the spatial and temporal change in uplift rates were not required while they provided a lot of evidence of the transient landscape. If there are other studies that propose a recent increase in uplift rates droved the transient response of the studied catchments, I suggest that you introduce those studies first and present your hypothesis.*

AR:  **Raised issue: uniform uplift** Apparently confusing statements about spatially and temporally constant uplift rates were mentioned as the third point. We think that this is just a minor misunderstanding.

**Performed task: revision of the introduction** We have revised and extended the introduction section and it should now be clear that region has been uplifted slowly since about 8 Myr. The term "basin inversion" is explained.

RC:  *The fourth issue is the contextualization. As a person who is unfamiliar with the geologic history of this region, it was hard to understand the significance of this study in terms of the evolution of the massif. Although the authors provide some references on geologic background of the massif, they did not clearly point out a research gap. Also, since the influence of rock strength on landscape evolution is well known, I suggest the authors explain more about the implication of their findings to put the current study in a wider context. I agree that bedrock strength strongly controls topography of the massif; however, the current study may end up with a case study without proper contextualization.*

AR: **Raised issue: contextualization** The reviewer criticizes the lack of contextualization.

AR: **Performed task:** We agree that there was a lack of contextualization and that we have left the reader in the dark at a few points. This applies in particular to the introduction, the results/discussion with the missing explanation of why we propose a lithological control without showing topographical metrics for contrasting lithological units, but also to the section describing the Aschach catchment. We are confident that we were able to solve these issues in the revised version by better linking the individual sections and placing them in a broader context.

**1.2. Line-by-line comments**

RC: *L47-50: Is the vertical velocity field based on GNSS observation?*

AR: The recent vertical velocity field is based on GNSS. The long term uplift was determined by burial ages from caves and the sedimentary record of the Molasse basin. We have revised the section.

RC: *L58–59: Influence of rock property on fluvial landscape is not exactly the same as those of uplift because the mechanisms of rock property and uplift control the erosional landscape are clearly different.*

AR: We are referring here to the channel slope at steady state and see no error in the statement. However, we have rephrased the sentence to make this clear. Furthermore, we have expanded the section to include the transient state and explain how a non-vertical contact can affect the geometry of the channel.

RC: *L82–83, 'Curiously, the drainage divide...': It wasn'y clear what you intended to say with this sentence.*
AR: *Okay right, we removed "curiously".*

RC: *L93: Please explain what the basin inversion is (e.g. spatial and temporal pattern of crustal deformation).*
AR: *In this context, basin inversion means the transition from a depositional environment to an eroding landscape due to uplift. We thought the term was clear, but we define what we mean by the term basin inversion.*

RC: *L106–108: In some catchment, granitic rock constitutes only a smaller part of the catchments (catchments at the western and eastern part of the study area). If the content of quartz is not similar among the rock types, resulting erosion rates are siginificantly biased towards rates in areas of specific rock type. I suggest explaining if the quartz content is similar among rock types examined.*

AR: You are right - thank you for the comment. We have added the information that all catchments predominantly consist of quartz-rich rocks (granites and gneisses). However, amphibolites and marbles (quartz-free) occur to a lesser extent in the eastern catchments (7, 8, 9).

RC: *L110 & Figure1: Please clarify the location of the physiographic transitions. This is a crucial information in this study; however, I could not understand where it was.*

AR: The physiographic transition represents the boundary between incised and low relief landscapes. The boundary follows the major rivers and their tributaries, and drawing the boundary as a line is not useful because then all you can see are just lines. We have therefore included a new figure (new Fig. 2) that shows the topography of the western part of the study region in perspective and thus gives a better impression of the landscape ad the contrasting landscape types. We also added a figure showing geophysical relief and normalised steepness index to the supplement. This should make the spatial occurrence of the two landscape types clearly visible. For those who want to get a very detailed impression, the supplement contains figures for each of the 20 catchments studied.

**RC:** *Fig1(a): I suggest using a continuous color scale rather than discrete scale, which helps to differeciate the elevated plateau and incised valley.*

**AR:** Done - However, we are not writing about plateaus but about low relief surfaces. Please see also new Fig. 2.

**RC:** *L139: Clarify 'pre-processing'.*

**AR:** The DEMs were preprocessed as described in the first paragraph of the Data and Methods section. We removed the term "pre-processing" to avoid confusion.

**RC:** *L153-154, 'The choice of…': This sentence would be incorrect. What matters in the choice of reference concavity is the gap between the concavity determined for each river profiles and the reference concavity.*

**AR:** Sorry, but the statement about $\theta_{\mathrm{ref}}$ is correct. The difference between the choices $\theta_{\mathrm{ref}} = 0.45$ and $0.5$ is only a factor of $A^{0.05}$ in $k_{\mathrm{sn}}$, regardless of the properties of the considered river.

**RC:** *L169–172: I do not think using elevation slices is helpful because you can manually separate the elevated plateau and incised valley.*

**AR:** Of course this could be done manually but it doesn't mean that another approach wouldn't also be useful are even better. In fact, we received exactly the opposite comment in another study – why go into the field or use a hand-drawn map at all when it can be done much more precisely using numerical methods. Expert-based mapping always means a subjective result and limited reproducibility , which is why we decided against it. In particular there are no plateaus but low relief surfaces and mapping of hundreds of isolated low relief surfaces is not only subject to poor reproducibility but can be replaced by an analysis of the relief, as we have done. We were searching for simple catchment-wide metrics that correlate well with catchment-wide erosion rates. Instead of expert-based mapping we rely on the area fraction below and above a certain geophysical relief threshold to discriminate between low relief and incised landscapes. Since we have seen that the lowest parts of the catchments show greater relief than the mid elevations, we still find the quartile approach useful as an additional topographic metric. In particular as the adjustment of channels from new local baselevels migrates upstream.

**RC:** *L180: You may want to move some parts in section 5.3 to here.*

**AR:** We are not fully convinced that "Model Setup" should be part of the Method section as region specific properties are explained. Anyhow, after implementing the conceptual model of landscape evolution (new Fig. 2) it makes sense shifting the Model Setup to the method section, which we have done. As described in the general points of criticism, however, we have relaxed the structure of the article and no longer follow a strict outline.

**RC:** *L201: How did you determine the spatial variation of precipitation?*

**AR:** This was obviously misunderstood. We use the precipiation module of OpenLEM to define the origin of a large river. We do not consider spatial variation in precipitation.

**RC:** *L205, result: I suggest rewriting this section to clarify the difference in gemorphc indices and erosion rates between the elevated plateau and incised valley or the difference between rock type.*

**AR:** This suggestion misses the objective of the study. However, we agree that the objectives of the study where not clear. We have revised the introduction and new implemented section 2 in order to explain the working hypothesis. In this study we present catchment-wide erosion rates that we correlate with catchment-wide topographic metrics (to be consistent). The catchments contain different proportions of incised topography and low relief topography. We address this with the area fraction of relief larger or smaller certain thresholds

(hillslopes) and the fraction of $k_{sn}$ larger or smaller certain thresholds (drainage system). With respect to lithology, we distinguish between sedimentary cover and crystalline basement according to our research hypothesis. These two rock types differ significantly in their erodibility but this does not mean that low relief and steep landscapes can be directly linked to the two constrasting rock types (please see revised introduction and new section 2 for explanation). We observe that after the erosion of the sedimentary cover, the low steepness of rivers is inherited and remains for a long time (response time). Thus, bedrock consisting of crystalline basement features steep and flat river segments (and everything in between). A separation according to lithology thus makes little sense.

The only remaining catchment where both rock types occur over a large area is the Aschach catchment. Based on this catchment, we describe how lithology controls the geometry of the landscape. In the revised version of this section it should be clear why topographic properties of the Aschach catchment explain the evolution of the entire region and comparable geological settings.

**RC:** *L260-262: Geomorphc indices of the catchment 10 do not seem to be 'exceptionally high'.*

AR: Okay, changed.

**RC:** *L308: I do not see arguments on the influence of rock properties in result section.*

AR: We have clarified that we compare catchments with crystalline bedrock. It should be clear now that we do not make any further distinction here between granites and high-grade metamorphic gneisses and migmatites. In this study, we only distinguish between cover and basement rocks.

**RC:** *L351–353: Did you find the difference in erosion rates using nested-catchment samples (Lainitz: P03, P04; Kleine Mühl: P17, P18)?*

AR: There are differences, but they are not significant. It would require a larger number of erosion rates in the course of a river. This is because low relief surfaces occur predominantly at medium altitudes and their influence is therefore difficult to isolate in the erosion signal.

**RC:** *L355–363: This paragraph reads like a geologic setting of the study area, which may be more appropriate in the section 2.*

AR: Right, we have transfered some of the geological detail to the introduction and deleted this paragraph.

**RC:** *L367–369: I could not understand this sentence. Does this sentence imply the northern catchment have already been adjusted to the contrast in rock stregnth?*

AR: Apart from a few remnants, the sedimentary cover has already been removed and the bedrock consists of crystalline basement. Since the response time for adaptation of the channel gradient in the hard bedrock is much greater than in the soft cover sediments, the adjustment is incomplete. This results in a bimodal landscape with steep rivers and slopes close to the receiving river and a topography with low gradients at mid-elevations. It should be clear after the revision.

**RC:** *L431–432: How did you determine K=1? Since K strongly controls the model outputs, it is important to justify your choice.*

AR: The selected value for $K = 1\ \mathrm{Myr}^{-1}$ is in the range of the values given in the literature for hard rock. We now provide a reference for granites. As we already wrote in the last version of the manuscript: *The long-term uplift rates of the region and erodibility of the different rocks are poorly constrained so that the presented model results only show one possible scenario for timing and rates of topography build-up in the Southern Bohemian Massif. Higher uplift rates would lead to a stronger expression of landscape bimodality. Smaller*

*bedrock erodibilities would have a similar effect as larger uplift rates. A further increase of the erodibility contrast from 1:10 between crystalline basement of the Bohemian Massif and the Neogene sediments of the Molasse Basin leads to similar topographic patterns with slightly more pronounced escarpments. With significantly smaller erodibility contrasts, the landscape bimodality is mainly due to fluvial prematurity and vanishes towards morphological equilibrium. However, the morphology of the Aschach catchment shows that both hillslopes and rivers are significantly steeper in the crystalline basement than in the Molasse sediments, supporting the assumption of pronounced differences of the bedrock in resistance to erosive surface processes.*

RC: *L435–436: Does the smoothed topography retain the bimodal landscape of the Bohemian Massif? I guess the shape of this smoothed surface strongly controls the final results. Showing something like DSM of the top of layer L2 maybe helpful.*

AR: As can be seen from the evolution of the topography in the Aschach catchment, the original thickness of the sediments (basin geometry) has a major influence on the topography and also in our model. This applies in particular to the formation of escarpments and elevated low relief surfaces. However, there are two length scales when we speak of a bimodal landscape. On the one hand, there is the difference between the Molasse basin (lowlands) and the Bohemian Massif (low mountain range) and, on the other hand, there are the steep, incised and low relief surfaces within the Bohemian Massif. The former is clearly determined by the geometry of the basin and shows different landscapes due to the contrasting bedrock properties (sediments versus crystalline basement), the latter shows the adaptation of the landscape to the hard rocks that were exposed after the erosion of the sedimentary cover and thus the response time. In both cases, however, it is not the case that an original surface reappears, but rather that different landscapes emerge due to the spatial and temporal change in the substrate properties. The surface of the hard basement can be found in the code supplement in the geotiff (BedRock_Model.tif) and can be analyzed in detail if interested. Anyhow It is not clear to us how we could implement your suggestion :"Showing something like DSM of the top of layer L2 maybe helpful"

RC: *L443–444: Again, why did you set Kd of L1 10 times as large as that of L2?*

AR: See author response to: L431–432

RC: *Figure8: I suggest presenting in a plan-view. Also, it maybe better to show the modeled area in a large-scale map such as in Figure1.*

AR: On this point, we do not agree. The model is used to illustrate how topographic patterns and erosion rates evolve in a landscape where soft sediments lie on a hard substrate, and not about creating a look-a-like od the study region. The focus is rather on the fundamental processes and landforms in the evolution of landscapes in such settings. The perspective view in combination with the videos (perspective view) in the supplement and with the new Figure 2 (perspective view) shows very clearly the emergence of a bimodal landscape under uniform uplift conditions.

**2. Review 2**

Reviewer 2 hast addressed two critical issues: (a) schematic outlining of the proposed model of landscape evolution and (b) additional statistics to show the uncertainties in the simple correlation between erosion rates and topographical metrics. We were very glad to take up the suggestions and quite pleased with how the article developed with these enhancements.

**2.1. General Comments**

**RC:** *In the paper from Robl et al., the authors use a combination of topographic analysis, cosmogenic erosion rates, and landscape evolution modeling to explore some of the nuances of a tectonically quiescent landscape with significant contrasts in erodibility and find that this erodibility contrast is likely critical in explaining the morphology of the landscape and the extent to which catchment averaged erosion rates do (or do not) follow sensible relations with the morphology. Overall I liked this paper and think it will make a strong and interesting contribution to ESurf after some revisions. Most of my comments reflect that while I feel like I eventually followed the authors arguments, it took a few more logical jumps than maybe were needed - and may have been unduly biased by my own perspectives. Either way, I think the paper could benefit from a bit more hand holding in terms of walking readers through the reason certain things are done and exactly what is being argued for. Most of this is addressed in my more line by line (or section by section) comments that follow, but I have two general comment to start with (one an actual comment and another more of a question). I look forward to seeing a version of this published sometime in the future.*

**AR:** Thank you very much for this constructive review, which helped to improve our study into a well-rounded publication. We implemented the suggestions and worked out the "red line" from the introduction to the discussion / conclusion more clearly ("it took a few more logical jumps than maybe were needed"). The additional references were indeed very helpful!

**RC:** *I think the paper could really benefit from a schematic outlining in simple form what is being argued for in the context of the lithologic contrast. In theory, the model outputs do this (Figure 8), and while these are useful to some extent in validating the authors proposed model, in practice it's actually hard to see what is actually happening through the detail. A simple cartoon would probably help. Along with what, it would help to emphasize (and I think is a point that is easily missed, unless I'm making it up for myself) I that the "bedrock barrier" as you describe it will persist for sometime after the overlying soft material is removed. I.e., at present, really only catchment 19 has the "setup" that you're invoking, but critical to the argument is that (presumably) many of these catchments had molasse "sitting" on them before, but it even though it has been stripped away, the morphological impact remains as the streams haven't fully adjusted yet. It's an important point that's pretty obvious from past work on lithologic contrasts in layered stratigraphy, but this is a great field example of it. Explaining this a bit more might help walk readers through the idea more and a simple cartoon would, in my opinion at least, really work well as a touchstone to do so.*

**AR:** **Performed task: Schematic outlining** We were happy to take up the suggestion and created a conceptual model of landscape evolution for a region with soft over hard rocks and uniform uplift. We have also written a new section on this (section 2), so that it is much easier for the reader to follow the article.

**RC:** *Assuming I haven't read way too much into what your data is showing and what you're interpreting from it, do you think there's more that could be done in terms of thinking about time? I.e., would parsing out which catchments (1) still have molasse, (2) don't have molasse but have preservation of the "bedrock barrier" as a morphological fingerprint of the molasse having been there but now stripped away, or (3) have effectively fully adjusted – and the spatial relationships of these catchments to each other – tell you anything about the overall response time of the system and the potential timescale over which the bedrock barrier can persist. In other words, how long after the stripping of the molasse will its morphological presence be felt? This is a question that you can answer from landscape evolution models, but the timescale will depend on the (appropriate) values of erodibility, diffusivity, scaling exponents, etc., where as you here have an interesting natural example to maybe tease it out. Maybe not possible, maybe too much for this paper, but seems like something fun to consider.*

**AR:** **System response time:** These are definitely exciting questions that we have discussed a lot over the last

year. The questions you raise are extremely exciting and important for our understanding of how landscapes evolve at the periphery of mountain ranges, where the crystalline basement is covered by weakly consolidated sediments of various thickness. However, to determine critical parameters of the stream power law ($K$, $n$) and make statements about the response time of the geomorphic system, we would require, among other things, a significantly higher number of measured erosion rates. We know that with the existing data set (20 erosion rates), we would be treading on very thin ice here. However, we are currently planning a follow-up project to address these and a series of other exiting research questions in the context of Old orogens - young topography. So we are well aware of these exciting questions, which however, go far beyond the scope of the actual paper. Anyhow, we would be happy to discuss some of the posed questions with you in the future.

**2.2. Line-by-line comments**

**RC:** *L157-160: I doubt the results would be substantially different, but is there a specific reason you're using geophysical relief as opposed to the more common local relief (e.g., DiBiase et al., 2010)?*

AR: Local vs geophyiscal relief: We don't think that using local relief instead of geophyiscal relief would change a lot as long as the window sizes are similar. We like the concept of geophysical relief in terms of the Davisian cycle of erosion. Assuming a uniform large-scale uplift, where rivers incise into the growing plateau from the rim, geophysical relief represents to the total erosion at each point in the landscape.

**RC:** *L163-170: I think you need to lay out a bit more rationale for why the quartile approach is useful or appropriate here. I know that you do briefly in the sentences that follow, but given where the paper ends up, I imagine that many would expect that a more logical way to parse the landscape analysis would be by lithology. Indeed, it seems like Reviewer 1 specifically suggests this as well. This may indeed be useful, but I suspect some part of why you're not doing this and are opting for the quartile approach instead is because your hypothesis and subsequent analysis suggests that there will be a good amount of morphological variability by lithologic unit (especially in the underlying hard unit) because of the formation of the "bedrock barrier". I think this is an important point to make, both because it helps to explain why a morphologic analysis filtered by lithology might not be useful in this instance (and thus helps to lay out the underlying rational for your methodology a bit more clearly) but also a general takeaway for others doing these types of analyses.*

AR: This is a very good point! Thank you! In the revised version, the effect of soft over hard rocks on topography development is presented in more detail. In general and with reference to the current literature, this is already mentioned in the introduction and specifically again in section 2 when describing the conceptual figure. We also explain in the revised version why it makes little sense to separate topographic metrics by different rocks types in such a geological setting.

**RC:** *L170-172: Probably a more nuanced point, but in terms of morphological responses to bedrock erodibility contrasts, these don't necessarily have to propagate up from a stable base level, i.e., depending on the channel slope and the orientation of the contact between bedrock with contrasting erodibility, these contrasts may appear virtually anywhere along a profile, specifically in cases when a contact broadly dips in the same direction as channel slope. This is seen in a variety of simulations of fluvial erosion through contrasting erodibilities (e.g., Forte et al., 2016; Wolpert & Forte, 2021).*

AR: Right - good point again! We have considered this comment in the introduction and in the new section 2. Great references by the way!

**RC:** *Section 4.1: There's nothing wrong with the analysis here per se (and it's certainly a very standard approach), but there's a case to be made that given that many of the sampled basins are clearly in a*

*transient state, simple regression of mean erosion rates vs mean topographic statistics would not really be expected to tell us much (and to the extent that they do show a pattern, there's reason to be suspicious of it as there's a good chance of it being spurious to some degree or another). For example, from looking at something like figure 5c compared to 5f, I suspect that if many of these values were plotted with their appropriate uncertainties, that it would be more clear to readers that the degree of correlation should be viewed with more skepticism, i.e., some of the basins have a mean ksn that is effectively the same as the standard deviation on that value. I wonder if the depth of the analysis here ends up being more confusing than it needs to be in the sense of where it seems like you are taking the reader compared to where we eventually end up. Simply put, most of these basins are not the type that are well described with a single mean value because of the morphologic variability, so that there is a quasi linear trend in these is maybe not actually very relevant.*

AR: We are aware that topographic and erosion rate variations **within transient catchments** can only be characterized to a limited extent using catchment-wide topographic metrics. We chose the approach of catchment-wide topographic metrics to be consistent with the catchment-wide erosion rates - currently the best way to determine erosion rates at a time scale relevant for landscape evolution. These values are only averages and underestimate the erosion rate for the steep incised parts of the landscape and overestimate it for the low relief areas. On average, they should be correct regardless of whether the catchment is in a steady state or in a transient state.

In the simplest detachment-limited model for fluvial erosion, the erosion rate is linearly proportional to the stream power ($A^\theta * S$) with $A$, $S$ and $\theta$ representing contributing drainage area, channel slope and the concavity index. Therefore, the catchment average of the erosion rate should correspond to the mean $k_{sn}$. However, to get around the problem with mean values, we investigated many other statistical measures and determined the area fractions where topographic metrics exceed or fall below threshold values, and split the catchments into elevation slices. Although the limited number of catchments means that the uncertainty is high, we are convinced that the correlations between erosion rate and topographic metrics are still meaningful. However, we fully agree that some additional statistics may be helpful to present and communicate the quality of the linear regressions. Please see revised figure 7 and accompanying text for explanation.

RC: *Figure 5: Per the above comment, it seems a bit disingenuous to show these plots without uncertainties. Even if these are not considered statistically in a meaningful way, at least visually readers could assess the extent to which a linear relationship is actually extractable. And similarly, a goodness of fit metric that includes the uncertainty (e.g., reduced chi squared, etc.) might be more relevant than a simple coefficient of determination.*

AR: We agree :-)! Please see revised figure 7 and accompanying text. We are confident that this provides a much better representation of the uncertainty of the approach.

RC: *Figure 7d: The yellow markers help, but it might be useful to also show where the contacts occur on the long profile (e.g., maybe colored bars marking which unit is exposed along the river).*

AR: We have revised the Figure accordingly. Indeed the figure benefited from the revision.

RC: *L400-4010: While simpler than your models in the sense that they only consider detachment limited erosion and obviously not being calibrated to the specific details of your study area (e.g., must faster rock uplift rates), these topographic and erosion pattern are also broadly observed in "soft over hard" models in Forte et al. (2016). Specifically, the development of the low gradient "bench" of underlying hard rocks during and after the stripping of the overlying softer rocks but also the depression of the erosion rates upstream of the contact until the profiles have responded and steepened (e.g., their Figures 2-4).*

AR:  Correct - our model is quite similar compared to the model of Forte et al. (2016). In the revised version we have implemented findings from Forte et al. (2016). Indeed this is a very relevant article for our study.

RC:  *Section 5.3.1: What values do you use for m and n? I don't think I saw these specified either in this section or the relevant methods section on the landscape models? As I'm sure the authors are aware, in simple stream power models at least, the value of n dictates a fair bit of the type of behavior in layered stratigraphy (e.g., Perne et al., 2017) so it's important to specify.*

AR:  We added the information on the values for m = 0.5 and n = 1 in the revised version (method section).

On behalf of the co-authors

Jörg Robl

**References**

Wetzlinger, K., Robl, J., Liebl, M., Dremel, F., Stüwe, K., and von Hagke, C.: Old orogen – young topography: Evidence for relief rejuvenation in the Bohemian Massif, Austrian Journal of Earth Sciences, 116, 17–38, , 2023.